# SketchBoost: Fast Gradient Boosted Decision Tree for Multioutput Problems

**Leonid Iosipoi**
Sber AI Lab and HSE University, Moscow, Russia
iosipoileonid@gmail.com

**Anton Vakhrushev**
Sber AI Lab, Moscow, Russia
btbpanda@gmail.com

## Abstract

Gradient Boosted Decision Tree (GBDT) is a widely-used machine learning algorithm that has been shown to achieve state-of-the-art results on many standard data science problems. We are interested in its application to multioutput problems when the output is highly multidimensional. Although there are highly effective GBDT implementations, their scalability to such problems is still unsatisfactory. In this paper, we propose novel methods aiming to accelerate the training process of GBDT in the multioutput scenario. The idea behind these methods lies in the approximate computation of a scoring function used to find the best split of decision trees. These methods are implemented in SketchBoost, which itself is integrated into our easily customizable Python-based GPU implementation of GBDT called Py-Boost. Our numerical study demonstrates that SketchBoost speeds up the training process of GBDT by up to over $40$ times while achieving comparable or even better performance.

## 1 Introduction

Gradient Boosted Decision Tree (GBDT) is one of the most powerful methods for solving prediction problems in both classification and regression domains. It is a dominant tool today in application domains where tabular data is abundant, for example, in e-commerce, financial, and retail industries. GBDT has contributed to a large amount of top solutions in benchmark competitions such as Kaggle. This makes GBDT a fundamental component in the modern data scientist's toolkit.

The main focus of this paper is the scalability of GBDT to multioutput problems. Such problems include multiclass classification (a classification task with more than two mutually exclusive classes), multilabel classification (a classification task with more than two classes that are not mutually exclusive), and multioutput regression (a regression task with a multivariate response variable). These problems arise in various areas such as Finance [Obermann and Waack, 2016], Multivariate Time Series Forecasting [Zhai, Yao, and Zhou, 2020], Recommender Systems [Jahrer, Töscher, and Legenstein, 2010], and others.

There are several extremely efficient, open-source, and production-ready implementations of gradient boosting such as XGBoost [Chen and Guestrin, 2016], LightGBM [Ke, Meng, Finley, Wang, Chen, Ma, Ye, and Liu, 2017], and Cat-

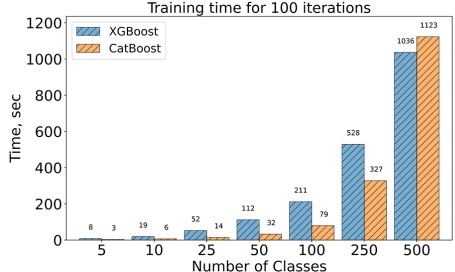

Figure 1: Training time of XGBoost and CatBoost for different number of classes on a synthetic dataset for multiclass classification. Synthetic dataset contains 2000k instances, each described by 100 features. The maximal tree depth was limited to 6. The experiment was conducted on GPU. Further details are given in the Supplementary Material.

36th Conference on Neural Information Processing Systems (NeurIPS 2022).

Boost [Prokhorenkova, Gusev, Vorobev, Dorogush, and Gulin, 2018]. Even for them, learning a GBDT model for moderately large datasets can require much time. Furthermore, this time also grows with the output size of a model. Figure 1 demonstrates how rapidly the training time of XGBoost and CatBoost grows with the output dimension. Consequently, the number of possible applications of GBDT in the multioutput regime is very limited.

GBDT is a boosting-based algorithm that ensembles decision trees as "base learners". At each boosting step, a newly added tree improves the ensemble by minimizing the error of an already built composition. There are two possible strategies on how to use GBDT to handle a multioutput problem.

- *One-versus-all strategy.* Here, at each boosting step, a single decision tree is built for every output. Consequently, every output is handled separately. XGBoost and LightGBM use this strategy.

- *Single-tree strategy.* Here, at each boosting step, a single multivariate decision tree is built for all outputs. Consequently, all outputs are handled together. CatBoost uses this strategy.

The computational complexity of both strategies is proportional to the number of outputs. Specifically, the one-versus-all strategy requires fitting a separate decision tree for each single output at each boosting step. The single-tree strategy requires scanning all the output dimensions (a) to estimate the information gain during the search of the best tree structure and (b) to compute leaf values of a decision tree with a given structure (see details in Section 2). A straightforward idea to reduce the training time of single-tree GBDT is to exclude some of the outputs during the search of the tree structure which is the most time-consuming step of GBDT. However, this turns out to be rather challenging since it is unclear what outputs contribute the most to the information gain. In this paper, we address this problem and propose novel methods for fast scoring of multivariate decision trees which show a significant decrease in computational overhead without compromising the performance of the final model.

**Related work.** Many suggestions have been made to speed up the training process of GBDT. Some methods reduce the number of data instances used to train each base learner. For example, Stochastic Gradient Boosting (SGB) [Friedman, 2002] chooses a random subset of data instances, gradient-based one-side sampling (GOSS) [Ke et al., 2017] keeps the instances with large gradients and randomly drops the instances with small gradients, and Minimal Variance Sampling (MVS) [Ibragimov and Gusev, 2019] randomly chooses the instances to maximize the estimation accuracy of split scoring. Similarly, some methods reduce the number of features. For example, one can choose a random subset of features or use principal component analysis or projection pursuit to exclude weak features; see [Jimenez and Landgrebe, 1999, Zhou, 2012, Appel et al., 2013]. LightGBM [Ke et al., 2017] uses exclusive feature bundling (EFB) where sparse features are greedily bundled together. CatBoost [Prokhorenkova et al., 2018] replaces categorical features with numerical ones using a special algorithm based on target statistics. Finally, some methods reduce the number of split candidates during the split scoring. The pre-sorted algorithm [Mehta et al., 1996] enumerates all possible split points on the pre-sorted feature values. The histogram-based algorithm [Alsabti et al., 1998, Jin and Agrawal, 2003b, Li et al., 2007] buckets continuous feature values into discrete bins and uses these bins to construct feature histograms.

Regarding the multioutput regime, the existing methods to accelerate the training process of GBDT naturally fall into the following two categories: problem transformation and algorithm adaptation. Transformation methods (see, for example, [Hsu et al., 2009, Tai and Lin, 2012, Kapoor et al., 2012, Cissé et al., 2012, Wicker et al., 2016]) reduce the number of targets before training a model. They mainly differ in the choice of compression and decompression techniques and significantly rely on the problem structure or data assumptions. These methods pay a price in terms of prediction accuracy due to the loss of information during the compression phase, and as a result, they do not consistently outperform the full baseline. Adaptation methods directly extend some specific algorithms to efficiently solve multioutput problems. To the best of our knowledge, there are only two algorithm adaptation works for GBDT. Namely, Si et al. [2017] and Zhang and Jung [2021] consider models with sparse output and discuss how to utilize this sparsity to enforce the leaf values to be also sparse. Their modifications of GBDT are called GBDT-Sparse and GBDT-MO (sparse).

We approach the problem of fast GBDT training in the multioutput regime from a different perspective. Namely, instead of employing the model sparsity, we, loosely speaking, approximate the scoring function used to find the best tree structure using the most essential outputs while keeping other boosting steps without change. The methods we suggest are completely different from the ones

mentioned above and can be applied to models with both dense and sparse outputs. Moreover, our methods can be easily combined with transformation methods (by compressing the outputs beforehand and decompressing predictions afterward) or the sparsity utilization as in GBDT-Sparse and GBDT-MO (by computing the optimal leaf values with sparsity constraint as in these algorithms).

**Contributions.**   The contributions of this work can be summarized as follows.

- We propose and theoretically justify three novel methods to speed up GBDT on multioutput tasks. These methods are generic, they can be used with any loss function and do not rely on any specific data assumptions (for example, sparsity or class hierarchy) or the problem structure (for example, multilabel or multiclass). Moreover, they do not drop down the model quality and can be easily integrated into any GBDT realization that uses the single-tree strategy.

- We implemented the proposed methods in SketchBoost. SketchBoost itself is a part of our Python-based implementation of GBDT called Py-Boost. This implementation seems to be of independent interest since it does not use low-level programming languages and is easily customizable. Although it is written in Python, it is fast since it works on GPU.

- We present an empirical study using public datasets which demonstrates that SketchBoost achieves comparable or even better performance compared to the existing state-of-the-art boosting toolkits but in remarkably less time.

**Paper Organization.**   First, we review the GBDT algorithm in Section 2. Next, we propose methods leading to a noticeable reduction in the training time of GBDT on multioutput tasks in Section 3. We illustrate the performance of these methods on real-world datasets in Section 4. Proofs and experiment details are postponed to the Supplementary Material.

## 2   Preliminaries

Let $\{(x_i, y_i)\}_{i=1}^n$ be a dataset with $n$ samples, where $x_i \in \mathbb{R}^m$ is an $m$ dimensional input and $y_i \in \mathbb{R}^d$ is a $d$ dimensional output. Let also $\mathcal{F}$ be a class of base learners, that is, functions $f : \mathbb{R}^m \to \mathbb{R}^d$. In Gradient Boosting, the idea of which goes back to Schapire [1990], Freund [1995], Freund and Schapire [1997], the model $F_T$ uses $T \in \mathbb{N}$ base learners $f \in \mathcal{F}$ and is trained in an additive and greedy manner. Namely, at the $t$-th iteration, a newly added base learner $f$ improves the quality of an already built model $F_{t-1}$ by minimization of some specified loss function $l : \mathbb{R}^d \times \mathbb{R}^d \to \mathbb{R}$,

$$\mathcal{L}_t(f) = \sum_{i=1}^n l(y_i, F_{t-1}(x_i) + f(x_i)).$$

This optimization problem is usually approached by the Newton method using the second-order approximation of the loss function

$$f_t^* \in \underset{f \in \mathcal{F}}{\operatorname{argmin}} \left\{ \sum_{i=1}^n \left( g_i^\top f(x_i) + \frac{1}{2} \big( f(x_i) \big)^\top H_i f(x_i) \right) + \Omega(f) \right\}, \tag{1}$$

where we omitted a term independent of $f$; here $\Omega(f)$ is a regularization term, usually added to build non-complex models, and

$$g_i = \nabla_a l(y, a) \Big|_{\substack{y = y_i \\ a = F_{t-1}(x_i)}}, \quad H_i = \nabla_{aa}^2 l(y, a) \Big|_{\substack{y = y_i \\ a = F_{t-1}(x_i)}}. \tag{2}$$

Due to the complexity of optimization over a large set of base learners $\mathcal{F}$, the problem (1) is solved typically in a greedy fashion which leads us to an approximate minimizer $f_t$. Finally, the model $F_t$ is updated by applying a learning rate $\varepsilon > 0$ typically treated as a hyperparameter: $F_t = F_{t-1} + \varepsilon f_t$.

GBDT uses decision trees as the base learners $\mathcal{F}$; see the seminal paper of Friedman [2001]. A decision tree is a model built by a recursive partition of the feature space into several disjoint regions. Each final leaf is assigned to a value, which is a response of the tree in the given region. Based on this construction mechanism, a decision tree $f$ can be expressed as

$$f(x) = \sum_{j=1}^J v_j \cdot [x \in R_j],$$

where [predicate] denotes the indicator function, $J$ is the number of leaves, $R_j$ is the $j$-th leaf, and $v_j \in \mathbb{R}^d$ is the value of $j$-th leaf. The problem of learning $f_t$ can be naturally divided into two separate problems: (1) finding the best tree structure (dividing the feature space into $J$ areas $R_1, \ldots, R_J$), and (2) fitting a decision tree with a given structure (computing leaf values $v_1, \ldots, v_J$).

**Finding the leaf values.**    Since decision trees take constant values at each leaf, for a decision tree $f_t$ with leaves $R_1, \ldots, R_J$, we can optimize the objective function from (1) for each leaf $R_j$ separately,

$$v_j = \operatorname*{argmin}_{v \in \mathbb{R}^d} \left\{ \sum_{x_i \in R_j} \left( g_i^\top v + \frac{1}{2} v^\top H_i v \right) + \frac{\lambda}{2} \|v\|^2 \right\} = - \left( \sum_{x_i \in R_j} H_i + \lambda I \right)^{-1} \left( \sum_{x_i \in R_j} g_i \right),$$

where we employ $l_2$ regularization on leaf values with a parameter $\lambda > 0$; here $I$ denotes the identity matrix and $\| \cdot \|$ denotes the Euclidean norm.

It is worth mentioning that if the loss function $l$ is separable with respect to different outputs, all Hessians $H_1, \ldots, H_n$ are diagonal. If it is not the case, it is a common practice to purposely simplify them to this extent in order to avoid time-consuming matrix inversion. It is done so in most of the single-tree GBDT algorithms (for example, CatBoost, GBDT-Sparse, and GBDT-MO). We will also follow this idea in our work. For diagonal Hessians, the optimal leaf values can be rewritten as

$$v_j = - \frac{\sum_{i \in R} g_i^j}{\sum_{i \in R} h_i^j + \lambda}, \quad \text{where } g_i = \begin{pmatrix} g_i^1 \\ \vdots \\ g_i^d \end{pmatrix} \text{ and } H_i = \begin{pmatrix} h_i^1 & \ldots & 0 \\ \vdots & \ddots & \vdots \\ 0 & \ldots & h_i^d \end{pmatrix}. \tag{3}$$

**Finding the tree structure.**    Substituting the leaf values from (3) back into the objective function, and omitting insignificant terms, we obtain

$$\text{Loss}(f_t) = - \frac{1}{2} \sum_{j=1}^{J} S(R_j), \quad \text{where} \quad S(R) = \sum_{j=1}^{d} \frac{\left( \sum_{x_i \in R} g_i^j \right)^2}{\sum_{x_i \in R} h_i^j + \lambda}. \tag{4}$$

The function $S(\cdot)$ will be referred to as the scoring function. To find the best tree structure, we use a greedy algorithm that starts from a single leaf and iteratively adds branches to the tree. At a general step, we want to split one of existing leaves. To do this, we iterate through all leaves, features, and thresholds for each feature (they are usually determined by the histogram-based algorithm). For all leaves $R$ and all possible splits for $R$, say $R_{\text{left}}$ and $R_{\text{right}}$, we compute the impurity score given by $S(R_{\text{left}}) + S(R_{\text{right}})$. The best split is considered the one which achieves the largest impurity score. This is equivalent to maximization of the information gain which is usually defined as the difference between values of the loss function before and after the split, that is,

$$\text{Gain} = -0.5 \Big( S(R) - \big( S(R_{\text{left}}) + S(R_{\text{right}}) \big) \Big).$$

Similar to the previous step, some simplifications can be made to speed up computation of the scoring function which is done a tremendous number of times. For instance, GBDT-Sparse does not use the second-order information at all (Hessians are simplified to identity matrices). In the multioutput regime of CatBoost, the second-order derivatives are left out during the split search and are used only to compute leaf values. GBDT-MO uses the second-order derivatives in both steps but it increases the computational complexity twice (histograms for both gradients and Hessians need to be accumulated).

## 3    Sketched Split Scoring

In this section, we propose three novel methods to speed up the split search for multivariate decision trees. These methods can achieve a good balance between reducing the computational complexity in the output dimension and keeping the accuracy for learned decision trees. They are generic and can be used together with the methods mentioned in the Related work section that aim at reducing the number of sample instances, features, or split candidates. Moreover, the proposed methods are easy to implement upon modern boosting frameworks such as XGBoost, LightGBM, and CatBoost.

As it was mentioned before, there are two "best practices" to speed up the training of a GBDT model on multioutput tasks: (a) to totally ignore the second-order derivatives during the split search and (b) to use only the main diagonal of the second-order derivatives to compute the leaf values. It is done so, for example, in CatBoost, one of the few boosting toolkits that use the single-tree strategy and achieve state-of-the-art results on multioutput problems. We will also develop our work on this basis.

The proposed methods are applied at each boosting step before the search for the best tree structure and after first- and second-order derivatives (see (2)) are computed. The key idea of the proposed

methods is to reduce the number of gradient values used in the split search so that the scoring function $S$ from (4) or, equivalently, the information gain will not change much. Specifically, the scoring function without the second-order information can be rewritten as

$$S_G(R) = \frac{\left\|G^\top v_R\right\|^2}{|R| + \lambda}, \quad \text{where } G = \begin{pmatrix} g_1^1 & g_1^2 & \cdots & g_1^d \\ \vdots & \vdots & \ddots & \vdots \\ g_n^1 & g_n^2 & \cdots & g_n^d \end{pmatrix} \text{ and } v_R = \begin{pmatrix} [x_1 \in R] \\ \vdots \\ [x_n \in R] \end{pmatrix}.$$

Here $G \in \mathbb{R}^{n \times d}$ is the gradient matrix and $v_R$ is the indicator vector of the leaf $R$ (its $i$-th coordinate is equal to 1 if $x_i \in R$ and 0 otherwise). Note that we added the subscript to $S$ to indicate its dependence on the gradient matrix $G$. To reduce the complexity of computing $S_G$ in $d$, we approximate it with $S_{G_k}$ for some other matrix $G_k \in \mathbb{R}^{n \times k}$ with $k \ll d$. We will refer to $G_k$ as the sketch matrix and to $k$ as the reduced dimension or sketching dimension. We emphasize that $G_k$ is assumed to be used only in building histograms and finding the tree structure. After this, the optimal leaf values of a tree are assumed to be computed fairly using the full gradient matrix $G$.

Further we discuss three novel methods to construct reasonably good sketches $G_k$ — Top Outputs, Random Sampling, and Random Projections. These methods are motivated by the minimization of the approximation error given by

$$\text{Error}(S_G, S_{G_k}) = \sup_R \left| S_G(R) - S_{G_k}(R) \right|.$$

Here the supremum is taken over all possible leaves $R$. The reason for this choice is that we want the proposed approximation to be universal and uniformly accurate for all splits we will possibly iterate over. In the Supplementary Material, we show that the proposed methods lead to a nearly-optimal upper bounds on the proposed error. Since the corresponding optimization problem is an instance of Integer Programming problem, methods leading to the optimal upper bounds can be obtained only by brute force, which is not an option in our case. For further details see the Supplementary Material.

## 3.1 Top Outputs

The key idea of Top Outputs is rather straightforward: to choose the columns of $G$ with the largest Euclidian norm. Namely, by a slight abuse of notation, let us denote the columns of $G$ by $g_1, \ldots, g_d$. Let also $i_1, \ldots, i_d$ be the indexes which sort the columns of $G$ in descending order by their norm, that is, $\|g_{i_1}\| \geq \|g_{i_2}\| \geq \ldots \geq \|g_{i_d}\|$. Now the full gradient matrix and its sketch can be written as

$$G = \begin{pmatrix} | & | & & | \\ g_1 & g_2 & \cdots & g_d \\ | & | & & | \end{pmatrix} \quad \text{and} \quad G_k = \begin{pmatrix} | & | & & | \\ g_{i_1} & g_{i_2} & \cdots & g_{i_k} \\ | & | & & | \end{pmatrix}.$$

The parameter $k$ here can be chosen adaptively to the norms of $g_1, \ldots, g_d$. We have not considered this generalization here since, in our view, it will greatly complicate the algorithm. Moreover, the adaptive choice of $k$ may result in large values for this parameter and hence less gain in training time.

It is worth pointing out that Top Outputs is akin to the Gradient-based One-Side Sampling (GOSS), which is successively used in LightGBM; see [Ke et al., 2017]. In GOSS, data instances with small gradients are excluded to speed up the split search. Similarly, Top Outputs excludes output components with small gradient values.

This method has one major drawback. This method chooses top $k$ output dimensions which may not vary much from step to step. For instance, if several columns have large norms and others have medium norms, Top Outputs may completely ignore the latter columns during the split search. Below we consider another method that deals with this problem by introducing the randomness in the choice of output dimensions.

## 3.2 Random Sampling

The probabilistic approach for algebraic computations, sometimes called the "Monte-Carlo method", is ubiquitous; we refer the reader to the monographs of Robert and Casella [2005],Mahoney [2011], and Woodruff [2014]. Here we consider its application to the fast split search.

The key idea of Random Sampling is to randomly sample the columns of $G$ with probabilities proportional to their norms. Namely, we define the sampling probabilities by

$$p_i = \|g_i\|^2 \Big/ \sum_{j=1}^d \|g_j\|^2, \quad i = 1, \ldots, d.$$

These probabilities are known to be optimal for random sampling since they minimize the variance of the resulting estimate; see, for example, [Robert and Casella, 2005]. Further, let $i_1, \ldots, i_k$ be independent and identically distributed random variables taking values $j$ with probabilities $p_j$, $j = 1, \ldots, k$. These random variables represent indexes of the chosen columns of $G$. Finally, we consider the following sketch

$$G_k = \begin{pmatrix} | & | & & | \\ \overline{g}_{i_1} & \overline{g}_{i_2} & \cdots & \overline{g}_{i_k} \\ | & | & & | \end{pmatrix}, \quad \text{where} \quad \overline{g}_i = \frac{1}{\sqrt{kp_i}} \, g_i.$$

The additional column normalization by $1/\sqrt{kp_i}$ is needed for unbiasedness of the resulting estimate.

There is a close affinity between Importance Sampling and Minimal Variance Sampling (MVS) of Ibragimov and Gusev [2019]. MVS decreases the number of sample instances in the split search by maximizing the estimation accuracy of split scoring. Our idea is the same with the only difference that it is applied to output dimensions rather than sample instances.

Random Sampling works well especially in the extreme cases as those mentioned above. For example, if several outputs have large weights and others have medium weights, Random Sampling will not ignore the latter outputs due to randomness. Or, if the number of outputs with large weights is larger than $k$, Random Sampling will choose different output dimensions at different steps. As a result, the corresponding base learners will also be quite different, which usually leads to a better generalization ability of the ensemble; see Breiman [1996].

### 3.3 Random Projections

In the previous section, the sketch $G_k$ was constructed by sampling columns from $G$ according to some probability distribution. This process can be viewed as multiplication of $G$ by a matrix $\Pi$, $G_k = G\Pi$, where $\Pi \in \mathbb{R}^{d \times k}$ has independent columns, and each column is all zero except for a 1 in a random location. In Random Projections, we consider sampling matrices $\Pi$, every entry of which is an independently sampled random variable. This results in using random linear combinations of columns of $G$ as columns of $G_k$.

This approach is based on the Johnson-Lindenstrauss (JL) lemma; see the seminal paper of Johnson and Lindenstrauss [1984]. They showed that projections $\Pi$ from $d$ dimensions onto a randomly chosen $k$-dimensional subspace do not distort the pairwise distances too much. Indyk and Motwani [1998] proved that to obtain the same guarantee, one can independently sample every entry of $\Pi$ using the normal distribution. In fact, this is true for many other distributions; see, for example, Achlioptas [2003]. Since there was no significant difference between distributions in our numerical experiments, we decided to focus on the normal distribution.

In Random Projections, we consider the following sketch

$$G_k = G\Pi,$$

where $\Pi \in \mathbb{R}^{d \times k}$ is a random matrix filled with independently sampled $\mathcal{N}(0, k^{-1})$ entries. In the Supplementary Material, we discuss why this choice leads to a nearly-optimal solution to the problem we consider and why the property of preserving the pairwise distances matters here.

Random Projections has the same merits as Random Sampling since it is also a random approach. Besides that, the sketch matrix $G_k$ here uses gradient information from all outputs since each column of $G_k$ is a linear combination of columns of $G$.

### 3.4 Complexity analysis.

Most of the GBDT frameworks use histogram-based algorithm to speed up split finding; see [Alsabti, Ranka, and Singh, 1998], [Jin and Agrawal, 2003a], and [Li, Wu, and Burges, 2008]. Instead of finding the split points on all possible feature values, histogram-based algorithm buckets feature values into discrete bins and uses these bins to construct feature histograms during training. Let us say that the number of possible splits per feature is limited to $h \ll n$ (usually $h \leq 256$ to store the histogram bin index using a single byte). It is shown in [Ke et al., 2017] that in the case of a single output, splitting a leaf $R$ with $n_R$ samples requires $O(mn_R)$ operations for histogram building and $O(hm)$ operations for split finding. As a result, if the actual tree construction is performed using a depth-first-search algorithm, the complexity of building a complete tree of depth $D$ is $O(Dnm + 2^D hm)$. In the

multioutput scenario, this complexity increases by $d$ times: splitting a leaf $R$ with $n_R$ samples costs $O(mn_R d + hmd)$ and depth-wise tree construction costs $O(Dmnd + 2^D hmd)$. The methods we propose reduce the impact of $d$ to $k$ with $k \ll d$. They require a preprocessing step which can be done, depending on the method, in $O(ndk)$ or $O(nd)$ operations. As a result, the complexity of building a complete tree of depth $D$ using the depth-first search can be reduced from $O(Dmnd + 2^D hmd)$ to $O(nd + Dmnk + 2^D hmk)$. Taking into account that $n$, $m$, and $d$ can be extremely large, these methods may lead to a significant improvement in the training time.

## 4 Numerical Experiments

In this section, we numerically compare (a) the proposed methods from Section 3 to speed up GBDT in the multioutput regime and (b) existing state-of-art boosting toolkits supporting multioutput tasks.

**Data.** The experiments are conducted on 9 real-world publicly available datasets from Kaggle, OpenML, and Mulan[1] for multiclass (4 datasets) and multilabel (3 datasets) classification and multitask regression (2 datasets). The associated details are given in the Supplementary Material.

**Py-Boost.** We implemented a simple and fast GBDT toolkit called Py-Boost. It is written in Python and hence is easily customizable. Py-Boost works only on GPU and uses Python GPU libraries such as CuPy and Numba. It follows the classic scheme described in [Chen and Guestrin, 2016]; further details are provided in the Supplementary Material. Py-Boost is available on GitHub[2].

**SketchBoost.** SketchBoost is a part of Py-Boost library which implements the following three sketching strategies for fast split search: **Top Outputs** (Section 3.1), **Random Sampling** (Section 3.2), and **Random Projections** (Section 3.3). For convenience, Py-Boost without any sketching strategy is referred to as **SketchBoost Full**. All the following experimental results and evaluation code are also available on GitHub[3].

**Baselines.** Primarily we compare SketchBoost with **XGBoost** (v1.6.0) and **CatBoost** (v1.0.5). There are two reasons why we have chosen these GBDT frameworks. First, they are commonly used among practitioners and represent two different approaches to multiouput tasks (one-vs-all and single-tree). Second, they can be efficiently trained on GPU, which allows us to compare their training time with GPU-based SketchBoost (with an exception for CatBoost which supports multilabel classification and multioutput regression tasks only on CPU). The reason why we have not considered LightGBM as a baseline is that it uses the same multiouput strategy as XGBoost (one-vs-all) and its latest version (v3.3.2) does not support multilabel classification and multioutput regression tasks without external wrappers. Further, we also compare SketchBoost with **TabNet** (v3.1.1), a popular deep learning model for tabular data; see [Arik and Pfister, 2021]. Our aim here is not to make an exhaustive comparison with existing deep learning approaches (it deserves its own investigation), but to make a comparison with a different in nature approach which moreover often has satisfactory complexity on large multioutput datasets.

**Experiment Design.** If there is no official train/test split, we randomly split the data into training and test sets with ratio 80%-20%. Then each algorithm is trained with 5-fold cross-validation (the train folds are used to fit a model and the validation fold is used for early stopping). We evaluate all the obtained models on the test set and get 5 scores for each model. The overall performance of algorithms is computed as an average score. As a performance measure, we use the cross-entropy for classification and RMSE for regression, but, for the sake of completeness, we also report the accuracy score for classification and R-squared score for regression in the Supplementary Material. For XGBoost, Catboost, and TabNet, we do the hyperparameter optimization using the Optuna framework [Akiba, Sano, Yanase, Ohta, and Koyama, 2019]. For SketchBoost, we use the same hyperparameters as for CatBoost (to speed up the experiment; we do not expect that hyperparameters will vary much since we use the same single-tree approach). The sketch size $k$ is iterated through the grid $\{1, 2, 5, 10, 20\}$ (or through a subset of this grid with values less than the output dimension). Further information on experiment design is given in the Supplementary Material.

---

[1] http://mulan.sourceforge.net/datasets.html
[2] https://github.com/sb-ai-lab/Py-Boost
[3] https://github.com/sb-ai-lab/SketchBoost-paper

Table 1: Test errors (cross-entropy for classification and RMSE for regression) ± their standard deviation.

| | SketchBoost | | | | Baseline | | |
|---|---|---|---|---|---|---|---|
| **Dataset** | **Top Outputs** (for the best $k$) | **Random Sampling** (for the best $k$) | **Random Projection** (for the best $k$) | **SketchBoost Full** (multioutput) | **CatBoost** (multioutput) | **XGBoost** (one-vs-all) | **TabNet** (multioutput) |
| **Multiclass classification** | | | | | | | |
| Otto (9 classes) | 0.4715 | 0.4636 | **0.4566** | 0.4697 | 0.4658 | 0.4599 | 0.5363 |
| | ±0.0035 | ±0.0026 | ±0.0023 | ±0.0030 | ±0.0033 | ±0.0028 | ±0.0063 |
| SF-Crime (39 classes) | 2.2070 | 2.2037 | 2.2038 | 2.2067 | **2.2036** | 2.2208 | 2.4819 |
| | ±0.0005 | ±0.0004 | ±0.0004 | ±0.0003 | ±0.0005 | ±0.0008 | ±0.0199 |
| Helena (100 classes) | 2.5923 | 2.5693 | **2.5673** | 2.5865 | 2.5698 | 2.5889 | 2.7197 |
| | ±0.0024 | ±0.0022 | ±0.0026 | ±0.0025 | ±0.0025 | ±0.0032 | ±0.0235 |
| Dionis (355 classes) | 0.3146 | 0.3040 | **0.2848** | 0.3114 | 0.3085 | 0.3502 | 0.4753 |
| | ±0.0011 | ±0.0014 | ±0.0012 | ±0.0009 | ±0.0010 | ±0.0020 | ±0.0126 |
| **Multilabel classification** | | | | | | | |
| Mediamill (101 labels) | 0.0745 | 0.0745 | **0.0743** | 0.0747 | 0.0754 | 0.0758 | 0.0859 |
| | ±1.3e-04 | ±1.3e-04 | ±1.1e-04 | ±1.3e-04 | ±1.1e-04 | ±1.1e-04 | ±3.3e-03 |
| MoA (206 labels) | 0.0163 | 0.0160 | **0.0160** | 0.0160 | 0.0161 | 0.0166 | 0.0193 |
| | ±2.2e-05 | ±1.0e-05 | ±6.0e-06 | ±9.0e-06 | ±2.6e-05 | ±2.1e-05 | ±3.0e-04 |
| Delicious (983 labels) | 0.0622 | 0.0619 | 0.0620 | 0.0619 | **0.0614** | 0.0620 | 0.0664 |
| | ±6.2e-05 | ±5.9e-05 | ±6.2e-05 | ±5.5e-05 | ±5.2e-05 | ±3.3e-05 | ±8.0e-04 |
| **Multitask regression** | | | | | | | |
| RF1 (8 tasks) | 1.1860 | 0.9944 | 0.9056 | 1.1687 | **0.8975** | 0.9250 | 3.7948 |
| | ±0.1366 | ±0.1015 | ±0.0582 | ±0.0835 | ±0.0384 | ±0.0307 | ±1.5935 |
| SCM20D (16 tasks) | 88.7442 | 86.2964 | **85.8061** | 91.0142 | 90.9814 | 89.1045 | 87.3655 |
| | ±0.6346 | ±0.4398 | ±0.5534 | ±0.3397 | ±0.3652 | ±0.4950 | ±1.3316 |

Table 2: Training time per fold in seconds.
(CatBoost does not support multilabel classification and multioutput regression tasks in the GPU mode.)

| | SketchBoost (GPU) | | | | Baseline (CPU/GPU) | | |
|---|---|---|---|---|---|---|---|
| **Dataset** | **Top Outputs** (for the best $k$) | **Random Sampling** (for the best $k$) | **Random Projection** (for the best $k$) | **SketchBoost Full** (multioutput) | **CatBoost** (multioutput) | **XGBoost** (one-vs-all) | **TabNet** (multioutput) |
| **Multiclass classification** | | | | | **GPU** | **GPU** | **GPU** |
| Otto (9 classes) | 113 | 102 | 89 | 131 | **73** | 1244 | 903 |
| SF-Crime (39 classes) | 705 | 676 | **612** | 1146 | 659 | 4016 | 2683 |
| Helena (100 classes) | 154 | 180 | **113** | 355 | 436 | 1036 | 1196 |
| Dionis (355 classes) | 1889 | 2038 | **419** | 23919 | 18600 | 18635 | 1853 |
| **Multilabel classification** | | | | | **CPU** | **GPU** | **GPU** |
| Mediamill (101 labels) | **251** | 263 | 294 | 1777 | 10164 | 2074 | 1231 |
| MoA (206 labels) | 103 | 189 | **87** | 696 | 9398 | 376 | 672 |
| Delicious (983 labels) | **575** | 664 | 1259 | 19553 | 20120 | 15795 | 2902 |
| **Multitask regression** | | | | | **CPU** | **GPU** | **GPU** |
| RF1 (8 tasks) | 369 | 396 | 340 | 413 | 804 | 315 | **207** |
| SCM20D (16 tasks) | 499 | 528 | 479 | 597 | 798 | 1432 | **296** |

**Results.** The final test errors are summarized in Table 1. Experiments show that, in general, SketchBoost with a sketching strategy obtains results comparable to or even better than the competing boosting frameworks. Promisingly, there is always a sketching strategy that outperforms SketchBoost Full. Random Projection achieves the best scores, but Random Sampling also performs quite well. The deterministic Top Outputs strategy scores less than other baselines everywhere. In addition, it is noticeable that the one-vs-all strategy implemented in XGBoost leads to a worse generalization ability than the single-tree strategy on most datasets.

The dependence of test scores on the sketch size $k$ for four datasets is shown in Figure 2; for other datasets see the Supplementary Material. It confirms the idea that, in general, the larger values $k$ we take, the better performance we obtain. Moreover, our numerical study shows that there is a wide range of values of $k$ for which sketching strategies work well; see the detailed results for all $k$ in the Supplementary Material. For most datasets, $k \leq 10$ is enough to obtain a result similar to or even better than SketchBoost Full or other baselines. Loosely speaking, an intuitive explanation of why reducing the output dimension may increase the ensemble quality is that building a tree using all outputs often leads to bad split choices for some particular outputs. Sketching strategies use small groups of outputs, which leads to better tree structures for these outputs and a more diverse ensemble overall. In this connection, the optimal value of $k$ strongly depends on the relations between the outputs in a given dataset. With limited resources in practice, we would recommend using a predefined value $k = 5$. It is common in GBDTs: modern toolkits have more than 100 hyperparameters, and many of them are not usually tuned (default values typically work well). But at the same time, one can always add $k$ to the set of hyperparameters that are tuned. In our view, an

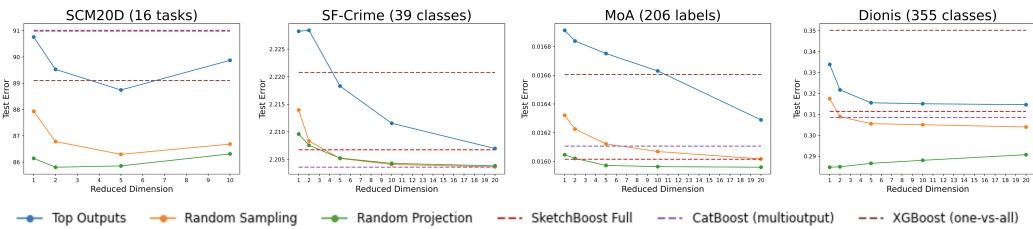

Figure 2: Dependence of test errors (cross-entropy for classification and RMSE for regression) on sketch dimension $k$.

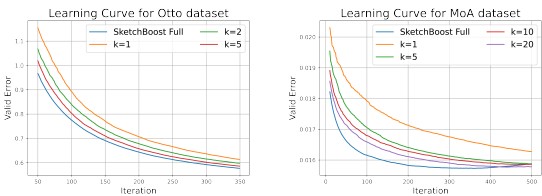

Figure 3: Learning curves for validation error for SketchBoost Full and SketchBoost with Random Sampling.

additional hyperparameter will not play a significant role here taking into account that hyperparameter optimization is usually done using the random search or Bayesian optimization.

Further, the learning curves for validation errors on some datasets are given in Figure 3. In general, it shows that small values of $k$ result in a slower error decay at early iterations. But if $k$ is properly defined, the validation error of SketchBoost with a sketching strategy is comparable to the error of SketchBoost Full, and hence both algorithms need approximately the same number of steps to convergence. This means that the proposed sketching strategies do not result in more complex models and do not significantly affect the model size or inference time. Detailed information on the number of steps to convergence for all strategies and baselines is given in the Supplementary Material.

SketchBoost does a good job in reducing the training time. In Table 2 we compare training times for SketchBoost, XGBoost, CatBoost, and TabNet. One can see that it significantly increases with the dataset size and, in particular, the output dimension. If a dataset is small, as, for example, RF1 (8 targets, 9k rows, 64 features) or Otto (9 classes, 61k rows, 93 features), our Python implementation is slightly slower than the efficient CatBoost or XGBoost GPU implementations written on low-level programming languages. But for Dionis (355 classes, 416k rows, 60 features), our implementation together with a sketching strategy becomes 40 times faster than XGBoost or CatBoost without sacrificing performance. Overall, we can conclude that the proposed sketching algorithms can significantly speed up SketchBoost Full and can lead to considerably faster training than other GBDT baselines. We recall that CatBoost can be trained on GPU only for multiclass classification tasks, and hence the time comparison with other algorithms on other tasks is not fair for CatBoost.

Finally, we see that all the GBDT implementations outperform TabNet in terms of test score on almost all tasks; see Table 1 again. These results confirm the conclusion from the recent surveys [Borisov, Leemann, Seßler, Haug, Pawelczyk, and Kasneci, 2021] and [Qin, Yan, Zhuang, Tay, Pasumarthi, Wang, Bendersky, and Najork, 2021] that algorithms based on gradient-boosted tree ensembles still mostly outperform deep learning models on tabular supervised learning tasks. Nevertheless, Table 2 shows that TabNet converges faster than GBDTs without sketching strategies. Moreover, TabNet is even faster than SketchBoost with sketching strategies on two regression tasks. The reason for this is that if the target dimension is high, it affects the complexity of a neural network only in the last layer and, in general, has little effect on the training time. Further, it is also worth mentioning that neural networks tend to have much more hyperparameters than GBDTs and, as the result, need more time to be properly fine-tuned. Further details on this experiment are given in the Supplementary Material.

**Comparison with GBDT-MO.** We also compare SketchBoost with GBDT-MO Full and GBDT-MO (sparse) from Zhang and Jung [2021] (we want to highlight that GBDT-Sparse from Si et al. [2017] does not have an open-source implementation). As sketching strategies, we consider here only Random Sampling and Random Projection. As the baseline, we consider only CatBoost on CPU (to make it comparable to GBDT-MO which works only on CPU). The datasets to compare and the best hyperparameters were taken from the original paper.

Table 3: Test scores (accuracy for classification and RMSE for regression) ± their standard deviation.

| | SketchBoost | | | GBDT-MO | | Baseline |
|---|---|---|---|---|---|---|
| **Dataset** | **Random Sampling** (for the best $k$) | **Random Projection** (for the best $k$) | **SketchBoost Full** (multioutput) | **GBDT-MO (sparse)** (for the best $k$) | **GBDT-MO Full** (multioutput) | **CatBoost** (multioutput) |
| **Multiclass classification** | | | | | | |
| MNIST (10 classes) | 0.9755 | 0.9740 | 0.9730 | 0.9758 | **0.9760** | 0.9684 |
| | ±0.0042 | ±0.0032 | ±0.0028 | ±0.0048 | ±0.0040 | ±0.0040 |
| Caltech (101 classes) | **0.5704** | 0.5623 | 0.5549 | 0.4796 | 0.4469 | 0.5049 |
| | ±0.0273 | ±0.0159 | ±0.0080 | ±0.0375 | ±0.0590 | ±0.0167 |
| **Multilabel classification** | | | | | | |
| NUS-WIDE (81 labels) | 0.9892 | **0.9897** | 0.9893 | 0.9892 | 0.9891 | 0.9893 |
| | ±0.0003 | ±0.0003 | ±0.0002 | ±0.0006 | ±0.0002 | ±0.0001 |
| **Multitask regression** | | | | | | |
| MNIST-REG (24 tasks) | 0.2661 | **0.2654** | 0.2660 | 0.2736 | 0.2723 | 0.2708 |
| | ±0.0019 | ±0.0012 | ±0.0019 | ±0.0017 | ±0.0026 | ±0.0023 |

Table 4: Training time per fold in seconds.

| | SketchBoost (GPU) | | | GBDT-MO (CPU) | | Baseline (CPU) |
|---|---|---|---|---|---|---|
| **Dataset** | **Random Sampling** (for the best $k$) | **Random Projection** (for the best $k$) | **SketchBoost Full** (multioutput) | **GBDT-MO (sparse)** (for the best $k$) | **GBDT-MO Full** (multioutput) | **CatBoost** (multioutput) |
| **Multiclass classification** | | | | | | |
| MNIST (10 classes) | 102 | 66 | **46** | 399 | 362 | 156 |
| Caltech (101 classes) | 15 | 16 | **13** | 1312 | 776 | 136 |
| **Multilabel classification** | | | | | | |
| NUS-WIDE (81 labels) | **36** | 72 | 87 | 3660 | 2606 | 13857 |
| **Multitask regression** | | | | | | |
| MNIST-REG (24 tasks) | 120 | **45** | 90 | 163 | 210 | 964 |

Summary results are presented in Table 3 and Table 4. SketchBoost with sketching strategies outperforms other algorithms on most datasets in terms of accuracy. GBDT-MO (sparse) is everywhere slower than GBDT-MO Full (because of optimization with a sparsity constraint). Furthermore, its training time is comparable to CatBoost. The time comparison with SketchBoost is not fair because of the GPU training, but, as it is shown, it is orders of magnitude faster. It is worth noting that SketchBoost Full is sometimes faster than SketchBoost with a sketching strategy. The reason for this is that if the dataset is small, then each boosting iteration requires little time. Therefore, when a sketching strategy is used, the speed up for each boosting iteration may be insignificant (especially because of ineffective utilization of GPU). At the same time, the number of iterations needed to convergence may be greater, which may result in an increase in the overall training time. Exactly this happened here. Further details on this experiment are given in the Supplementary Material.

## 5 Conclusion

In this paper, we presented effective methods to speed up GBDT on multioutput tasks. These methods are generic and can be easily integrated into any single-tree GBDT realization. On real-world datasets, these methods achieve comparable and sometimes even better results to the existing state-of-the-art GBDT implementations but in remarkably less time. The proposed methods are implemented in SketchBoost which itself is a part of our Python-based implementation of GBDT called Py-Boost. Figure 4 concludes this paper by showing the gain in training time of SkechBoost in the same experiment as in Figure 1 from the Introduction.

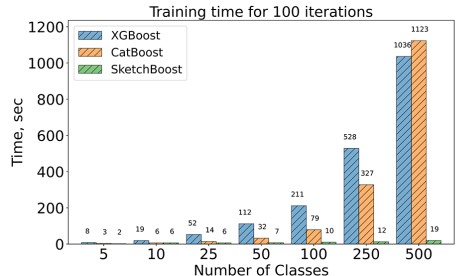

Figure 4: Training time of XGBoost, CatBoost, and SkechBoost in the same experiment as in Figure 1. Here SketchBoost uses Random Projection with sketch dimension $k = 5$. Further details are given in the Supplementary Material.

## Acknowledgements

We would like to thank Gleb Gusev and Bulat Ibragimov for helpful discussions and feedback for an earlier draft of this work, Dmitry Simakov and Mikhail Kuznetsov for the help with the TabNet experiments, and Maxim Savchenko and all the Sber AI Lab team for their support and active interest in this project. We would also like to thank the anonymous reviewers for their thoughtful feedback.

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
