# SketchBoost: Fast Gradient Boosted Decision Tree for Multioutput Problems, Supplementary Material

**Leonid Iosipoi**
Sber AI Lab and HSE University, Moscow, Russia
iosipoileonid@gmail.com

**Anton Vakhrushev**
Sber AI Lab, Moscow, Russia
btbpanda@gmail.com

## 1  Additional Information on Sketched Split Scoring Methods.

This section provides additional information on sketched split scoring methods from Section 3. Let us recall that the scoring function $S(R)$ for a leaf $R$ is given by

$$S_G(R) = \frac{\|G^\top v_R\|^2}{|R| + \lambda}, \quad \text{where } G = \begin{pmatrix} g_1^1 & g_1^2 & \cdots & g_1^d \\ \vdots & \vdots & \ddots & \vdots \\ g_n^1 & g_n^2 & \cdots & g_n^d \end{pmatrix} \text{ and } v_R = \begin{pmatrix} [x_1 \in R] \\ \vdots \\ [x_n \in R] \end{pmatrix}.$$

Here $G \in \mathbb{R}^{n \times d}$ is the matrix whose $i$-th row consists of gradient values $(g_i^1, \ldots, g_i^d) = \nabla_a l(y_i, a)|_{a = F_{t-1}(x_i)}$, $i = 1, \ldots, n$, and $v_R$ is the indicator vector of leaf $R$ (its $i$-th component equals 1 if $x_i \in R$ and 0 otherwise). To reduce the complexity of computing $S_G(R)$ in $d$, we approximate $S_G(R)$ with $S_{G_k}(R)$ for some sketch matrix $G_k \in \mathbb{R}^{n \times k}$ with $k \ll d$. The error of this approximation is measured by

$$\text{Error}(S_G, S_{G_k}) = \sup_R |S_G(R) - S_{G_k}(R)|. \tag{1}$$

The supremum here is taken over all possible leaves $R$, so that we aim at the optimization of the worst case. The reason for this is that we want the proposed approximation to be universal and uniformly accurate for all splits we will possibly iterate over.

Given the two matrices $G$ and $G_k$, the optimization problem from (1) is an instance of Integer Programming problem and hence is NP-complete. To obtain a closed-form solution, one needs to iterate through all possible leaves $R$ (that is, through all possible vectors $v_R$ with 0/1 entries). Since the brute force is not an option in our case, we will replace this problem with a relaxed one and will look for nearly-optimal solutions. We will show that reasonably good upper bounds on the error are obtained when $GG^\top$ is well approximated with $G_k G_k^\top$ in the operator norm; see Lemma 1.1. This observation links our problem to Approximate Matrix Multiplication (AMM). In the next sections, we review some deterministic and random methods from AMM and apply them to construct nearly optimal sketches $G_k$.

**Auxiliary Lemma.**  First let us state an auxiliary lemma which bounds the approximation error from (1) with the distance between $GG^\top$ and $G_k G_k^\top$ is the operator norm.

**Lemma 1.1.** *Let $G \in \mathbb{R}^{n \times d}$ and $G_k \in \mathbb{R}^{n \times k}$ be any two matrices. Then*

$$\text{Error}(S_G, S_{G_k}) \leq \|GG^\top - G_k G_k^\top\|. \tag{2}$$

*Proof.*  A direct computation yields

$$\sup_R |S_G(R) - S_{G_k}(R)| = \sup_R \left| \frac{\|G^\top v_R\|^2 - \|G_k^\top v_R\|^2}{|R| + \lambda} \right|$$

$$\leq \sup_R \frac{\|GG^\top - G_k G_k^\top\| \|v_R\|^2}{|R| + \lambda}.$$

Since $\lambda > 0$ and $\|v_R\|^2 \leq |R|$ ($v_R$ has $|R|$ non-zero entries equal to 1), the assertion follows. $\quad\square$

Note that in practice we do not need to compute $GG^\top$. Lemma 1.1 only provides a theoretical bound which will be used further. This bound is universal for all possible leafs $R$ and involves only the gradient matrix $G$ and its sketch $G_k$.

## 1.1 Truncated SVD

We start with the Truncated SVD algorithm since, by the matrix approximation lemma (the Eckart-Young-Mirsky theorem), it provides the optimal deterministic solution to AMM. The following proposition summarizes its performance.

**Proposition 1.2.** *Let $G \in \mathbb{R}^{n \times d}$ be any matrix. Let also $G_k \in \mathbb{R}^{n \times k}$ be the best $k$-rank approximation of $G$ provided by the Truncated SVD. Then*

$$\mathrm{Error}(S_G, S_{G_k}) \leq \sigma_{k+1}^2(G),$$

*where $\sigma_{k+1}^2(G)$ is $(k+1)$ largest singular value of $G$.*

*Proof.* Let $G = U\Sigma V^\top$ be the full SVD of $G$. Let also $G_k = U_k \Sigma_k$ be the $k$-rank Truncated SVD of $G$ where we keep only largest $k$ singular values and corresponding columns in $U$. Using Lemma 1.1, we get

$$\sup_R |S_G(R) - S_{G_k}(R)| \leq \|GG^\top - G_k G_k^\top\| = \|U\Sigma^2 U^\top - U_k \Sigma_k^2 U_k^\top\|.$$

Now the Eckart–Young–Mirsky theorem (for the spectral norm) yelds

$$\|U\Sigma^2 U^\top - U_k \Sigma_k^2 U_k^\top\| = \sigma_{s+1}^2(G),$$

which finishes the proof. $\quad\square$

This proposition asserts that to speed up the split search, the gradient matrix $G$ with $d$ columns can be replaced by its Truncated SVD estimate with $k$ columns. As a result, the scoring function $S_G$ will not change significantly provided that $(k+1)$ largest singular value of $G$ is small. The parameter $k$ here can be chosen adaptively depending of the spectrum of $G$ and values on $S_G$.

We haven't discussed Truncated SVD in the paper due to its computational complexity which is $O(\min\{nd^2, n^2d\})$; see [Golub and Van Loan, 1996]. As it was discussed in the Introduction, the computational complexity of GBDT scales linearly in the output dimension $d$. Consequently, the application of Truncated SVD will only increase this complexity. Further, we discuss methods with less computational costs.

## 1.2 Top Outputs

Top Outputs is a straightforward method which constructs the sketch $G_k$ by keeping $k$ columns of the gradient matrix $G$ with the largest Euclidian norm.

**Proposition 1.3.** *Let $G \in \mathbb{R}^{n \times d}$ be any matrix. Let also $G_k \in \mathbb{R}^{n \times k}$ be the sketch of $G$ given by Top Outputs. Then*

$$\mathrm{Error}(S_G, S_{G_k}) \leq \sum_{j=k+1}^{d} \|g_{i_j}\|^2.$$

*Proof.* Lemma 1.1 implies that

$$\sup_R |S_G(R) - S_{G_k}(R)| \leq \|GG^\top - G_k G_k^\top\|.$$

We rewrite $GG^\top$ and $G_k G_k^\top$ as an outer product of their columns,

$$GG^\top = \sum_{i=1}^{d} g_i g_i^\top \quad \text{and} \quad G_k G_k^\top = \sum_{j=1}^{k} g_{i_j} g_{i_j}^\top.$$

see Section 3.2 for notation. By construction, we have

$$\left\| GG^\top - G_k G_k^\top \right\| = \left\| \sum_{j=k+1}^{d} g_{i_j} g_{i_j}^\top \right\| \le \sum_{j=k+1}^{d} \left\| g_{i_j} \right\|^2,$$

and the proof is complete. □

This proposition shows that the approximation error is small when we cut out columns of $G$ with a small norm. Top Outputs method is less preferable than Truncated SVD in terms of the approximation error. Nevertheless, here the sketch $G_k$ can be computed in time $O(nd)$ which is linear in $d$ contrary to the Truncated SVD.

## 1.3  Random Sampling

In Random Sampling, we sample columns of $G$ according to probabilities proportional to their Euclidian norm. Before we proceed, let us denote the stable rank of $G$ by

$$\mathrm{sr}(G) = \frac{\|G\|_{\mathrm{Fr}}^2}{\|G\|^2},$$

where $\|\cdot\|$ denotes the spectral norm. The stable rank is a relaxation of the exact notion of rank. Indeed, one always has $\mathrm{sr}(G) \le \mathrm{rank}(G)$. But as opposed to the exact rank, it is stable under small perturbations of the matrix. Both exact and stable ranks are usually referred to as the intrinsic dimensionality of a matrix (in data-driven applications matrices tend to have small ranks).

**Proposition 1.4.** *Let $G \in \mathbb{R}^{n \times d}$ be any matrix. Let also $G_k \in \mathbb{R}^{n \times k}$ be a sketch obtained by Random Sampling. Then for any $\delta \in (0, 1)$, with probability at least $1 - \delta$,*

$$\mathrm{Error}(S_G, S_{G_k}) \le C_{G,\delta} \frac{\|G\|^2}{\sqrt{k}},$$

*where $C_{G,\delta}$ is a constant depending on $G$ and $\delta$ and is given by*

$$C_{G,\delta} = 2\sqrt{\mathrm{sr}(G) \log\left(\frac{4\,\mathrm{sr}(G)}{\delta}\right)}.$$

*Proof.* Lemma 1.1 yields

$$\sup_R \left| S_G(R) - S_{G_k}(R) \right| \le \left\| GG^\top - G_k G_k^\top \right\|.$$

Using Theorem 4.2 from [Holodnak and Ipsen, 2015], we obtain that for any $\varepsilon, \delta \in (0, 1)$ with probability at least $1 - \delta$,

$$\left\| GG^\top - G_k G_k^\top \right\| \le \varepsilon \|G\|^2$$

provided that $k \ge 3\,\mathrm{sr}(G) \ln(\frac{4r}{\delta})/\varepsilon^2$. Solving the latter inequality with respect to $\varepsilon$, we establish the assertion. □

This proposition states that the approximation error is, with high probability, of order $\|G\|^2/\sqrt{k}$ when the stable rank of $G$ is small. There is no definite answer whether this bound is better than the bounds obtained for other methods. The answer depends on the spectrum of $G$. Moreover, this bound is of probabilistic nature. Nevertheless, Random Sampling has the computational complexity $O(nd)$ and is as fast as Top Outputs.

## 1.4  Random Projections

Random Projections samples $k$ random linear combinations of columns of G to construct $G_k$.

**Proposition 1.5.** *Let $G \in \mathbb{R}^{n \times d}$ be any matrix. Let also $\Pi \in \mathbb{R}^{d \times k}$ be a random matrix filled with independently sampled $\mathcal{N}(0, k^{-1})$ entries. Set $G_k = G\Pi$. Then for any $\delta \in (0, 1)$, with probability at least $1 - \delta$,*

$$\mathrm{Error}(S_G, S_{G_k}) \le C_{G,\delta} \frac{\|G\|^2}{\sqrt{k}},$$

*where $C_{G,\delta}$ is a constant depending on $G$ and $\delta$ and is given by*

$$C_{G,\delta} = c\sqrt{\mathrm{sr}(G) + \ln\left(\frac{1}{\delta}\right)}.$$

*for some absolute constant $c > 0$.*

*Proof.* Using Lemma 1.1, we get

$$\sup_R \left|S_G(R) - S_{G_k}(R)\right| \leq \left\|GG^\top - G_k G_k^\top\right\|.$$

Now Theorem 1 from [Kyrillidis et al., 2014] implies that for any $\varepsilon, \delta \in (0, 1)$ with probability at least $1 - \delta$,

$$\|GG^\top - G_k G_k^\top\| \leq \varepsilon \|G\|^2$$

provided that $k \geq c\left(\mathrm{sr}(G) + \ln\ln(\frac{1}{\varepsilon}) + \ln(\frac{1}{\delta})\right)/\varepsilon^2$. If we set

$$\varepsilon = c'\sqrt{(\mathrm{sr}(G) + \ln(\frac{1}{\delta}))/k}$$

for another absolute constant $c'$, the assertion follows. $\qquad\square$

Comparing to Proposition 1.4, this bound is slightly better in terms of $C_{G,\delta}$. But the sketch $G_k$ here can be computed only in time $O(ndk)$ since it requires multiplication of $G$ and $\Pi$. To speed up it, one can use Fast JL transform [Ailon and Chazelle, 2009] or Sparse JL transform [Dasgupta, Kumar, and Sarlos, 2010], [Kane and Nelson, 2014].

## 2  Experiment Details

We remind the reader that our Python-based GPU implementation of GBDT called Py-Boost is available on GitHub[1]. The code to reproduce the experiments is also available on GitHub[2].

### 2.1  About Py-Boost

As was mentioned in the original paper, Py-Boost is written in Python and follows the classic scheme described in [Chen and Guestrin, 2016]. Meanwhile, it is a simplified version of gradient boosting, and hence it has a few limitations. Some of these limitations have been made to speed up computations, some — to remove unnecessary for our purposes features presented in modern gradient boosting toolkits (for example, categorical data handling). The complete list of these limitations is the following. Py-Boost supports: (a) computations only on GPU, (b) only the depth-wise tree growth policy, (c) only numeric features (with possibly NaN values), and (d) only histogram algorithm for split search (maximum number of bins for each feature is limited to 256).

Py-Boost uses GPU Python libraries such as CuPy, Numba, and CuML to speed up computations. XGBoost and CatBoost frameworks are also evaluated in the GPU mode where possible (CatBoost is evaluated on CPU on multilabel classification and multioutput regression tasks since it does not support them on GPU).

### 2.2  Experiment design

In our numerical experiments, we compare SketchBoost Full, SketchBoost with sketching strategies (Top Outputs, Random Sampling, Random Projections), XGBoost (v1.6.0) which uses the one-vs-all strategy, and CatBoost (v1.0.5) which uses the single-tree strategy, and a popular deep learning model for tabular data TabNet (v3.1.1). In this section, we discuss experiment design for GBDTs; details for TabNet are given in Section 2.4 below. The experiments are conducted on 9 real-world publicly available datasets from Kaggle, OpenML, and Mulan website for multiclass/multilabel classification and multitask regression. Datasets details are given in Table 1.

---

[1]`https://github.com/sb-ai-lab/Py-Boost`
[2]`https://github.com/sb-ai-lab/SketchBoost-paper`

Table 1: Dataset statistics.

| Dataset | Task | Rows | Features | Classes/Labels/Targets | Source | Download |
|---|---|---|---|---|---|---|
| Otto | multiclass | 61 878 | 93 | 9 | Kaggle | Link |
| SF-Crime | multiclass | 878 049 | 10 | 39 | Kaggle | Link |
| Helena | multiclass | 65 196 | 27 | 100 | OpenML | Link |
| Dionis | multiclass | 416 188 | 60 | 355 | OpenML | Link |
| Mediamill | multilabel | 43 907 | 120 | 101 | Mulan | Link |
| MoA | multilabel | 23 814 | 876 | 206 | Kaggle | Link |
| Delicious | multilabel | 16 105 | 500 | 983 | Mulan | Link |
| RF1 | multitask | 9125 | 64 | 8 | Mulan | Link |
| SCM20D | multitask | 8966 | 61 | 16 | Mulan | Link |

We remind the reader that if there is no official train/test split, we split the data into train and test sets with ratio 80%-20%. Datasets taken from Kaggle have the official train/test split, but since the platform hides the test set, we split the train test into the new train and test sets. Some of the datasets required data preprocessing since they contained categorical and datetime features (they cannot be handled by all of the GBDT implementations on the fly). The code for data preprocessing step is also provided in the Supplementary Material.

Experiments are performed on the server under OS Ubuntu 18.04 with 4 NVidia Tesla V100 32 GB GPUs, 48 cores CPU Intel(R) Xeon(R) Platinum 8168 CPU @ 2.70GHz and 386 GB RAM. We run all the tasks on 8 CPU threads and single GPU if needed.

The experiments are divided into two the following parts:

- **Parameter tuning.** Here we optimize hyperparameters for XGBoost and CatBoost. For Sketch-Boost we use the same hyperparameters as for CatBoost (to speed up the parameter tuning process; we do not expect that hyperparameters will vary much since we use the same tree building strategy). At this stage, parameters are estimated by 5-fold cross-validation using only the train set.

- **Model evaluation.** After the best parameters are found, we refit all the models using a longer training time. The models are trained again by 5-fold cross-validation, but their quality is estimated on the test set. The final score and time metrics are computed as the average of 5 values given by 5 models from the cross-validation loop.

During the parameter tuning, we use a slightly different setup (higher learning rate and less maximum number of rounds) to evaluate more trials and find a better hyperparameter set. Table 2 provides the associate details.

Table 2: Setup for the parameter tuning and model evaluation stages.

| Stage | Learning Rate | Max Number of Rounds | Early Stopping Rounds | Quality Estimation |
|---|---|---|---|---|
| Parameter tuning | 0.05 | 5000 | 200 | cross-validation |
| Models evaluation | 0.015 | 20000 | 500 | test set |

Since all the models are trained using cross-validation, the optimal number of boosting rounds is determined adaptively by early stopping on the validation fold. In this setup, the test set is used only for quality evaluation. As the primary quality measure, we use the cross-entropy for classification and RMSE for regression. But for the sake of completeness, we also report the accuracy for classification and R-squared for regression, see Section 2.5.

## 2.3 Hyperparameter tuning

It is quite challenging to perform a fair test among all the frameworks. Evaluation results depend not only on the sketching method or the strategy used to handle a multioutput problem (one-vs-all or single-tree), but also on hyperparameter setting and implementation details. To make our comparison as fair as possible, we perform a hyperparameter optimization for XGBoost and CatBoost using

the Optuna[3] framework that performs a sequential model-based optimization by the Tree-structured Parzen Estimator (TPE) method; see [Bergstra, Bardenet, Bengio, and Kégl, 2011]. The list of optimized hyperparameters is given in Table 3.

Table 3: Optimized hyperparameters.

| Parameter | Framework | Type | Default value | Search space | Log |
|---|---|---|---|---|---|
| Maximal Tree Depth | All | Int | 6 | [3:12] | False |
| Min Child Weight | XGBoost | Float | 1e-5 | [1e-5:10] | True |
| Min Data In Leaf | CatBoost | Int | 1 | [1: 100] | True |
| L2 leaf regularization | All | Float | 1 | [0.1: 50] | True |
| Rows Sampling Rate | All | Float | 1.0 | [0.5: 1] | False |

**Note**: Minimal data in leaf and Minimal child weight both control the leaf size but in different frameworks. It is the reason why they have different scales in the table.

The optimal number of rounds is determined by early stopping for the fixed learning rate; see Section 2.2 for the details. There are some other parameters of GBDT that are not optimized since not all the frameworks allow to change them. For example, the columns sampling rate cannot be changed in CatBoost in the GPU mode, and only the depth-wise grow policy is supported in SketchBoost.

The tuning process is organized in the following way:

- **Safe run.** First, we perform a single run with a set of parameters that are listed in Table 3 as default. These hyperparameters are close to the framework's default settings and can be considered as a safe option in the case when a good set of parameters is not found in the time limit.

- **TPE search.** Then we perform 30 iterations of parameters search with the Optuna framework. As it was mentioned, the training time may be quite long for the multioutput tasks, so we limit the search time to 24 hours in order to finish it in a reasonable amount of time.

- **Selection.** Finally, we determine the best parameters as the ones that achieve the best performance between all trials (both safe and TPE).

The best hyperparameters and the number of successful (finished in time) trials are given in Table 4.

## 2.4 Details on TabNet

Here we provide additional details on TabNet [Arik and Pfister, 2021]. We consider PyTorch (v1.7.0)-based implementation pytorch-tabnet library (v3.1.1). The evaluation is performed on the same datasets, under the same experiment setup including train/validation/test splits, and using the same hardware as described in Section 2.2. TabNet hyperparameters are taken from the original paper [Arik and Pfister, 2021] except the learning rate, batch size, and the number of epochs for train and early stopping. These hyperparameters are tuned in the following way:

- The optimal learning rate is tuned using the Optuna framework from the range $(1e - 5; 1e - 1)$ in the log scale in the same way at it is described in Section 2.3 (one single safe run with learning rate $2e - 2$ and then TPE search for 10 trials with time limit of 24 hours). The best trial is selected based on the cross-validation score and then it is evaluated on the test set.

- The optimal number of epochs is selected via early stopping with 16 epochs with no improve. Maximal number of epochs is limited to 500.

- The batch size is selected depending on the dataset size: 256 for data with less than 50k rows, 512 for 50-100k rows, and 1024 for more then 100k rows.

The proposed values are quite typical for tabular data according to [Gorishniy, Rubachev, Khrulkov, and Babenko, 2021], [Fiedler, 2021], and [Arik and Pfister, 2021]. The unsupervised pretrain proposed in the paper is also used on each cross-validation iteration. The selected hyperparameters are listed in the Table 5 below.

---

[3] https://github.com/optuna/optuna

Table 4: Results for hyperparameter optimization.

| Dataset | Framework | Min data in leaf | Min child weight | Rows sampling rate | Max depth | L2 leaf regularization | Completed trials |
|---------|-----------|------------------|------------------|--------------------|-----------|------------------------|------------------|
| Otto | CatBoost | 47 | – | 0.89 | 10 | 3.83 | 31 |
| | XGBoost | – | 0.00001 | 0.58 | 12 | 23.57 | 31 |
| SF-Crime | CatBoost | 2 | – | 0.94 | 11 | 2.65 | 31 |
| | XGBoost | – | 1.25682 | 0.92 | 12 | 8.37 | 24 |
| Helena | CatBoost | 2 | – | 0.55 | 6 | 12.77 | 31 |
| | XGBoost | – | 0.33734 | 0.50 | 6 | 23.33 | 31 |
| Dionis | CatBoost | 1 | – | 1.0 | 6 | 1.0 | 6 |
| | XGBoost | – | 0.00001 | 1.0 | 6 | 1.0 | 3 |
| Mediamill | CatBoost | 10 | – | 0.76 | 8 | 11.16 | 8 |
| | XGBoost | – | 1.90469 | 0.68 | 12 | 39.30 | 31 |
| MoA | CatBoost | 30 | – | 0.88 | 4 | 1.03 | 10 |
| | XGBoost | – | 0.00342 | 0.93 | 3 | 0.37 | 31 |
| Delicious | CatBoost | 76 | – | 0.62 | 12 | 13.05 | 5 |
| | XGBoost | – | 0.06235 | 0.72 | 11 | 33.22 | 10 |
| RF1 | CatBoost | 1 | – | 0.85 | 10 | 9.96 | 31 |
| | XGBoost | – | 0.00431 | 0.68 | 5 | 12.22 | 31 |
| SCM20D | CatBoost | 5 | – | 0.99 | 12 | 6.57 | 31 |
| | XGBoost | – | 9.93770 | 0.80 | 6 | 1.44 | 31 |

Table 5: Results for hyperparameter optimization for TabNet.

| Dataset | Learning rate | Batch size | Completed trials |
|---------|---------------|------------|------------------|
| Otto | 0.0394 | 512 | 11 |
| SF-Crime | 0.0200 | 1024 | 3 |
| Helena | 0.0080 | 512 | 9 |
| Dionis | 0.0110 | 1024 | 7 |
| Mediamill | 0.0315 | 256 | 11 |
| MoA | 0.0200 | 256 | 10 |
| Delicious | 0.0231 | 256 | 11 |
| RF1 | 0.0200 | 256 | 11 |
| SCM20D | 0.0200 | 256 | 11 |

## 2.5 Additional experimental results

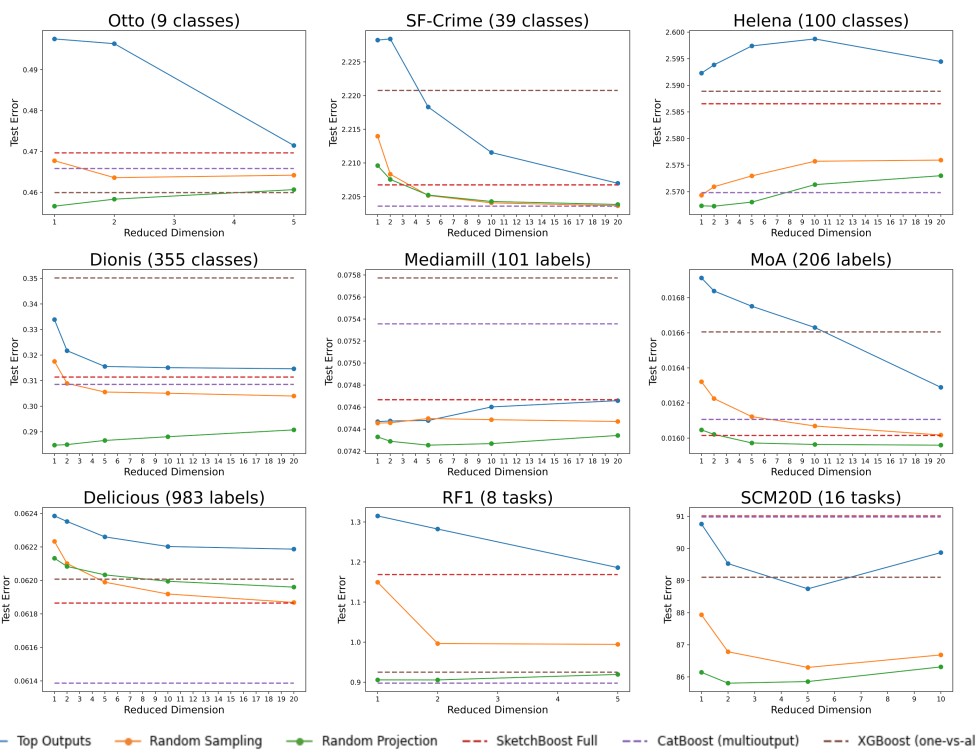

Figure 1: Dependence of test errors (cross-entropy for classification and RMSE for regression) on sketch dimension $k$ for all datasets.

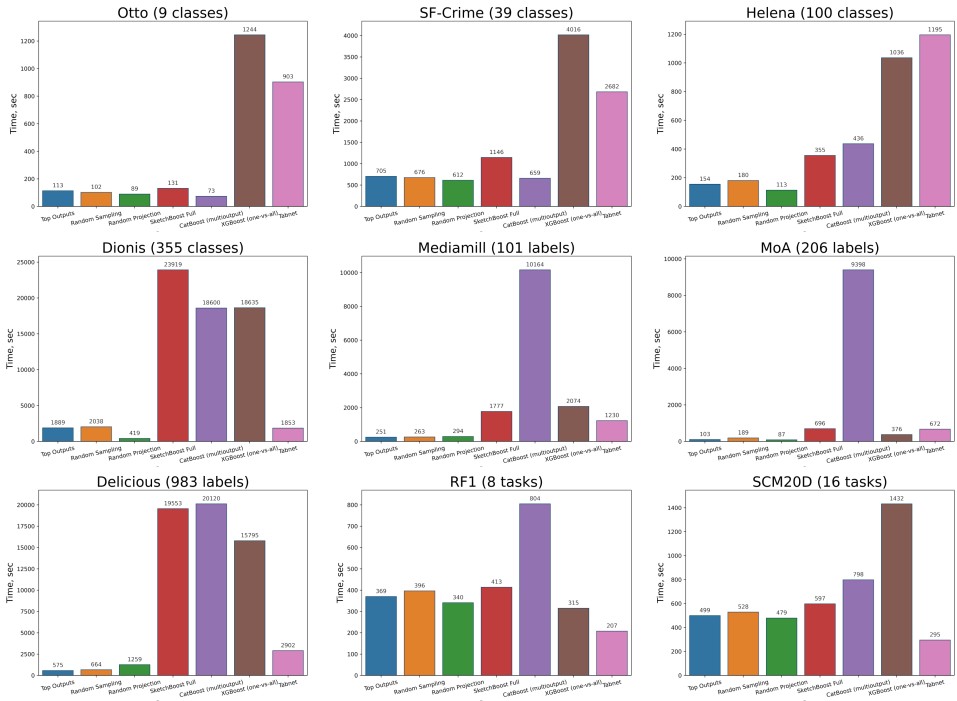

Figure 2: Training time per fold in seconds for the best sketching dimension $k$.

Table 6: Test errors (cross-entropy for classification and RMSE for regression) $\pm$ their standard deviation.

| | | Dataset | | | | | | | | |
|---|---|---|---|---|---|---|---|---|---|---|
| Algorithm | | Otto | SF-Crime | Helena | Dionis | Mediamill | MoA | Delicious | RF1 | SCM20D |
| XGBoost | | 0.4599 | 2.2208 | 2.5889 | 0.3502 | 0.0758 | 0.0166 | 0.0620 | 0.9250 | 89.1045 |
| | | ±0.0027 | ±0.0008 | ±0.0031 | ±0.0019 | ±1.1e-04 | ±2.1e-05 | ±3.3e-05 | ±0.0307 | ±0.4950 |
| CatBoost | | 0.4658 | 2.2036 | 2.5698 | 0.3085 | 0.0754 | 0.0161 | 0.0614 | 0.8975 | 90.9814 |
| | | ±0.0032 | ±0.0005 | ±0.0025 | ±0.0010 | ±1.1e-04 | ±2.6e-05 | ±5.2e-05 | ±0.0383 | ±0.3652 |
| TabNet | | 0.5363 | 2.4819 | 2.7197 | 0.4753 | 0.0859 | 0.0193 | 0.0664 | 3.7948 | 87.3655 |
| | | ±0.0063 | ±0.0199 | ±0.0235 | ±0.0126 | ±3.3e-03 | ±3.0e-04 | ±8.0e-04 | ±1.5935 | ±1.3316 |
| SketchBoost Full | | 0.4697 | 2.2067 | 2.5865 | 0.3114 | 0.0747 | 0.0160 | 0.0619 | 1.1687 | 91.0142 |
| | | ±0.0029 | ±0.0003 | ±0.0025 | ±0.0008 | ±1.3e-04 | ±9.0e-06 | ±5.5e-05 | ±0.0835 | ±0.3396 |
| Top Outputs | $k=1$ | 0.4975 | 2.2282 | 2.5923 | 0.3339 | 0.0745 | 0.0169 | 0.0624 | 1.3151 | 90.7613 |
| | | ±0.0030 | ±0.0004 | ±0.0024 | ±0.0017 | ±1.3e-04 | ±4.1e-05 | ±6.3e-05 | ±0.0721 | ±0.3988 |
| | $k=2$ | 0.4964 | 2.2284 | 2.5938 | 0.3217 | 0.0745 | 0.0168 | 0.0624 | 1.2823 | 89.5284 |
| | | ±0.0041 | ±0.0001 | ±0.0025 | ±0.0019 | ±1.4e-04 | ±2.0e-05 | ±5.1e-05 | ±0.1363 | ±0.8352 |
| | $k=5$ | 0.4715 | 2.2183 | 2.5974 | 0.3155 | 0.0745 | 0.0168 | 0.0623 | 1.1860 | 88.7442 |
| | | ±0.0035 | ±0.0005 | ±0.0018 | ±0.0013 | ±1.1e-04 | ±1.6e-05 | ±6.3e-05 | ±0.1365 | ±0.6345 |
| | $k=10$ | – | 2.2116 | 2.5987 | 0.3151 | 0.0746 | 0.01660 | 0.0622 | – | 89.8727 |
| | | – | ±0.0025 | ±0.0019 | ±0.0014 | ±1.0e-04 | ±2.5e-05 | ±5.5e-05 | – | ±0.3126 |
| | $k=20$ | – | 2.2070 | 2.5945 | 0.3146 | 0.0747 | 0.0163 | 0.0622 | – | – |
| | | – | ±0.0005 | ±0.0020 | ±0.0010 | ±1.1e-04 | ±2.2e-05 | ±6.2e-05 | – | – |
| Random Sampling | $k=1$ | 0.4677 | 2.2140 | 2.5693 | 0.3175 | 0.0745 | 0.0163 | 0.0622 | 1.1495 | 87.9358 |
| | | ±0.0019 | ±0.0003 | ±0.0022 | ±0.0009 | ±1.3e-04 | ±2.0e-05 | ±6.4e-05 | ±0.0674 | ±0.4111 |
| | $k=2$ | 0.4636 | 2.2083 | 2.5710 | 0.3089 | 0.0745 | 0.0162 | 0.0621 | 0.9965 | 86.7842 |
| | | ±0.0025 | ±0.0003 | ±0.0032 | ±0.0012 | ±9.1e-05 | ±1.5e-05 | ±5.5e-05 | ±0.1011 | ±0.5546 |
| | $k=5$ | 0.4642 | 2.2052 | 2.5730 | 0.3055 | 0.0745 | 0.0161 | 0.0620 | 0.9944 | 86.2964 |
| | | ±0.0020 | ±0.0005 | ±0.0024 | ±0.0012 | ±9.8e-05 | ±1.5e-05 | ±5.6e-05 | ±0.1014 | ±0.4398 |
| | $k=10$ | – | 2.2041 | 2.5757 | 0.3051 | 0.0745 | 0.0161 | 0.0619 | – | 86.6865 |
| | | – | ±0.0005 | ±0.0018 | ±0.0009 | ±1.0e-04 | ±1.5e-05 | ±5.2e-05 | – | ±0.2829 |
| | $k=20$ | – | 2.2037 | 2.5759 | 0.3040 | 0.0745 | 0.0160 | 0.0619 | – | – |
| | | – | ±0.0004 | ±0.0023 | ±0.0013 | ±1.1e-04 | ±1.0e-05 | ±5.9e-05 | – | – |
| Random Projection | $k=1$ | 0.4566 | 2.2096 | 2.5674 | 0.2848 | 0.0743 | 0.0160 | 0.0621 | 0.9058 | 86.1442 |
| | | ±0.0023 | ±0.0005 | ±0.0039 | ±0.0012 | ±1.1e-04 | ±1.0e-05 | ±5.9e-05 | ±0.0442 | ±0.4824 |
| | $k=2$ | 0.4583 | 2.2076 | 2.5673 | 0.2850 | 0.0743 | 0.0160 | 0.0621 | 0.9056 | 85.8061 |
| | | ±0.0028 | ±0.0006 | ±0.0026 | ±0.0011 | ±1.2e-04 | ±2e-05 | ±5.9e-05 | ±0.0581 | ±0.5533 |
| | $k=5$ | 0.4607 | 2.2052 | 2.5681 | 0.2866 | 0.0743 | 0.0160 | 0.0620 | 0.9193 | 85.8565 |
| | | ±0.0031 | ±0.0006 | ±0.0019 | ±0.0013 | ±1.2e-04 | ±1.9e-05 | ±5.6e-05 | ±0.0781 | ±0.3116 |
| | $k=10$ | – | 2.2043 | 2.5713 | 0.2881 | 0.0743 | 0.0160 | 0.0620 | - | 86.3126 |
| | | – | ±0.0003 | ±0.0024 | ±0.0010 | ±1.1e-04 | ±1.3e-05 | ±5.9e-05 | - | ±0.3710 |
| | $k=20$ | – | 2.2038 | 2.5730 | 0.2907 | 0.0743 | 0.0160 | 0.062 | – | – |
| | | – | ±0.0004 | ±0.0030 | ±0.0009 | ±1.2e-04 | ±6.0e-06 | ±6.2e-05 | – | – |

Table 7: Test errors (accuracy for classification and R-squared for regression) $\pm$ their standard deviation.

| | | Dataset | | | | | | | | |
|---|---|---|---|---|---|---|---|---|---|---|
| Algorithm | | Otto | SF-Crime | Helena | Dionis | Mediamill | MoA | Delicious | RF1 | SCM20D |
| XGBoost | | 0.8238 | 0.3326 | 0.3770 | 0.9193 | 0.9746 | 0.9971 | 0.9826 | 0.9997 | 0.9257 |
| | | ±0.0010 | ±0.0003 | ±0.0012 | ±0.0007 | ±4.3e-05 | ±8.0e-06 | ±6.0e-06 | ±3.2e-05 | ±0.0007 |
| CatBoost | | 0.8213 | 0.3352 | 0.3808 | 0.9234 | 0.9744 | 0.9971 | 0.9825 | 0.9997 | 0.9224 |
| | | ±0.0012 | ±0.0008 | ±0.0017 | ±0.0003 | ±7.6e-05 | ±5.0e-06 | ±1.7e-05 | ±4.1e-05 | ±0.0006 |
| TabNet | | 0.7972 | 0.2550 | 0.3503 | 0.8936 | 0.9709 | 0.9967 | 0.9816 | 0.9932 | 0.9281 |
| | | ±0.0030 | ±0.0037 | ±0.0060 | ±0.0032 | ±1.0e-03 | ±5.3e-05 | ±9.5e-05 | ±0.0037 | ±0.0022 |
| SketchBoost Full | | 0.8223 | 0.3343 | 0.3783 | 0.9227 | 0.9747 | 0.9971 | 0.9824 | 0.9995 | 0.9224 |
| | | ±0.0021 | ±0.0007 | ±0.0011 | ±0.0004 | ±6.8e-05 | ±6.0e-06 | ±1.4e-05 | ±5.5e-05 | ±0.0007 |
| Top Outputs | $k=1$ | 0.8172 | 0.3275 | 0.3773 | 0.9192 | 0.9747 | 0.9970 | 0.9823 | 0.9992 | 0.9228 |
| | | ±0.0022 | ±0.0006 | ±0.0014 | ±0.0006 | ±4.9e-05 | ±7.0e-06 | ±2.0e-05 | ±6.1e-05 | ±0.0006 |
| | $k=2$ | 0.8171 | 0.3275 | 0.3772 | 0.9214 | 0.9747 | 0.9970 | 0.9823 | 0.9993 | 0.9249 |
| | | ±0.0016 | ±0.0008 | ±0.0021 | ±0.0004 | ±4.7e-05 | ±4.0e-06 | ±2.0e-05 | ±1.2e-04 | ±0.0013 |
| | $k=5$ | 0.8210 | 0.3315 | 0.3760 | 0.9229 | 0.9747 | 0.9970 | 0.9823 | 0.9994 | 0.9262 |
| | | ±0.0016 | ±0.0003 | ±0.0019 | ±0.0004 | ±5.8e-05 | ±1.2e-05 | ±1.9e-05 | ±9.2e-05 | ±0.0010 |
| | $k=10$ | – | 0.3333 | 0.3757 | 0.9229 | 0.9747 | 0.997 | 0.9824 | – | 0.9243 |
| | | – | ±0.0003 | ±0.0013 | ±0.0003 | ±2.8e-05 | ±5.0e-06 | ±1.5e-05 | – | ±0.0005 |
| | $k=20$ | – | 0.3349 | 0.3766 | 0.9227 | 0.9747 | 0.9971 | 0.9824 | – | – |
| | | – | ±0.0006 | ±0.0008 | ±0.0007 | ±6.5e-05 | ±9.0e-06 | ±1.6e-05 | – | – |
| Random Sampling | $k=1$ | 0.8228 | 0.3320 | 0.3821 | 0.9224 | 0.9746 | 0.9971 | 0.9824 | 0.9995 | 0.9276 |
| | | ±0.0011 | ±0.0002 | ±0.0009 | ±0.0005 | ±5.6e-05 | ±5e-06 | ±2.0e-05 | ±5.4e-05 | ±0.0007 |
| | $k=2$ | 0.8236 | 0.3338 | 0.3827 | 0.9243 | 0.9746 | 0.9971 | 0.9824 | 0.9996 | 0.9293 |
| | | ±0.0025 | ±0.0003 | ±0.0015 | ±0.0003 | ±4.8e-05 | ±5.0e-06 | ±8.0e-06 | ±6.3e-05 | ±0.0009 |
| | $k=5$ | 0.8231 | 0.3348 | 0.3795 | 0.9250 | 0.9746 | 0.9971 | 0.9824 | 0.9996 | 0.9301 |
| | | ±0.0018 | ±0.0002 | ±0.0022 | ±0.0002 | ±4.0e-05 | ±1.2e-05 | ±1.3e-05 | ±6.0e-05 | ±0.0007 |
| | $k=10$ | – | 0.3351 | 0.3793 | 0.9250 | 0.9746 | 0.9972 | 0.9824 | – | 0.9295 |
| | | – | ±0.0004 | ±0.0018 | ±0.0002 | ±4.1e-05 | ±1.2e-05 | ±8.0e-06 | – | ±0.0004 |
| | $k=20$ | – | 0.3353 | 0.3795 | 0.9251 | 0.9746 | 0.9972 | 0.9824 | – | – |
| | | – | ±0.0005 | ±0.0017 | ±0.0005 | ±5.6e-05 | ±9.0e-06 | ±1.7e-05 | – | – |
| Random Projection | $k=1$ | 0.8258 | 0.3338 | 0.3834 | 0.9285 | 0.9747 | 0.9971 | 0.9824 | 0.9997 | 0.9304 |
| | | ±0.0010 | ±0.0003 | ±0.0033 | ±0.0004 | ±1.7e-05 | ±6.0e-06 | ±1.6e-05 | ±3.7e-05 | ±0.0008 |
| | $k=2$ | 0.8255 | 0.3344 | 0.3836 | 0.9287 | 0.9748 | 0.9971 | 0.9824 | 0.9997 | 0.9309 |
| | | ±0.0023 | ±0.0005 | ±0.0019 | ±0.0002 | ±5.6e-05 | ±3.0e-06 | ±1.8e-05 | ±4.4e-05 | ±0.0008 |
| | $k=5$ | 0.8251 | 0.3351 | 0.3835 | 0.9281 | 0.9748 | 0.9971 | 0.9824 | 0.9997 | 0.9308 |
| | | ±0.0023 | ±0.0005 | ±0.0018 | ±0.0002 | ±4.0e-05 | ±1.1e-05 | ±7.0e-06 | ±5.4e-05 | ±0.0005 |
| | $k=10$ | – | 0.3355 | 0.3824 | 0.9279 | 0.9748 | 0.9971 | 0.9824 | - | 0.9301 |
| | | – | ±0.0003 | ±0.0022 | ±0.0002 | ±4.0e-05 | ±6.0e-06 | ±9.0e-06 | - | ±0.0006 |
| | $k=20$ | – | 0.3357 | 0.3805 | 0.9275 | 0.9747 | 0.9971 | 0.9824 | – | – |
| | | – | ±0.0004 | ±0.0025 | ±0.0002 | ±2.9e-05 | ±5.0e-06 | ±1.2e-05 | – | – |

Table 8: Training time per fold in seconds for all sketching dimensions $k$.

| Algorithm | | Otto | SF-Crime | Helena | Dionis | Mediamill | MoA | Delicious | RF1 | SCM20D |
|---|---|---|---|---|---|---|---|---|---|---|
| | | | | | | **Dataset** | | | | |
| XGBoost | | 1244 | 4016 | 1036 | 18635 | 2074 | 376 | 15795 | 315 | 1432 |
| CatBoost | | 73 | 659 | 436 | 18600 | 10164 | 9398 | 20120 | 804 | 798 |
| TabNet | | 903 | 2563 | 1196 | 1853 | 1231 | 672 | 2902 | 207 | 296 |
| SketchBoost Full | | 131 | 1146 | 355 | 23919 | 1777 | 696 | 19553 | 413 | 597 |
| Top Outputs | $k = 1$ | 129 | 174 | 154 | 783 | 251 | 40 | 213 | 351 | 458 |
| | $k = 2$ | 126 | 207 | 151 | 810 | 276 | 45 | 229 | 364 | 476 |
| | $k = 5$ | 113 | 270 | 146 | 1003 | 313 | 59 | 274 | 369 | 499 |
| | $k = 10$ | – | 425 | 138 | 1293 | 386 | 69 | 375 | – | 551 |
| | $k = 20$ | – | 705 | 156 | 1889 | 529 | 103 | 575 | – | – |
| Random Sampling | $k = 1$ | 104 | 198 | 180 | 835 | 263 | 61 | 230 | 347 | 485 |
| | $k = 2$ | 102 | 219 | 180 | 880 | 273 | 75 | 243 | 354 | 491 |
| | $k = 5$ | 116 | 299 | 185 | 1087 | 319 | 104 | 314 | 396 | 528 |
| | $k = 10$ | – | 422 | 198 | 1404 | 399 | 135 | 432 | – | 590 |
| | $k = 20$ | – | 676 | 213 | 2038 | 559 | 189 | 664 | – | – |
| Random Projection | $k = 1$ | 89 | 136 | 109 | 419 | 235 | 26 | 212 | 331 | 466 |
| | $k = 2$ | 87 | 159 | 113 | 464 | 243 | 29 | 235 | 340 | 479 |
| | $k = 5$ | 107 | 233 | 116 | 629 | 294 | 39 | 295 | 393 | 528 |
| | $k = 10$ | – | 365 | 128 | 895 | 369 | 55 | 436 | – | 594 |
| | $k = 20$ | – | 612 | 149 | 1417 | 527 | 87 | 1259 | – | – |

Table 9: Number of boosting iterations to convergence (for GBDTs).
(Although the number of iterations for XGBoost is small, it uses the one-vs-all strategy,
and therefore the actual amount of trees in the ensemble equals this number multiplied by the output size $d$.)

| Dataset | SketchBoost | | | | Baseline | |
|---|---|---|---|---|---|---|
| | **Top Outputs**
(for the best $k$) | **Random Sampling**
(for the best $k$) | **Random Projection**
(for the best $k$) | **SketchBoost Full**
(multioutput) | **CatBoost**
(multioutput) | **XGBoost**
(one-vs-all) |
| **Multiclass classification** | | | | | | |
| Otto (9 classes) | 4799 | 5424 | 5201 | 4424 | 5534 | 2142 |
| SF-Crime (39 classes) | 3790 | 3726 | 3611 | 3754 | 3993 | 1212 |
| Helena (100 classes) | 15042 | 16975 | 11670 | 13492 | 11238 | 1563 |
| Dionis (355 classes) | 17039 | 17990 | 11509 | 18519 | 19858 | 2681 |
| **Multilabel classification** | | | | | | |
| Mediamill (101 labels) | 18623 | 19961 | 17826 | 17927 | 8983 | 1878 |
| MoA (206 labels) | 2606 | 5542 | 2093 | 2240 | 4239 | 471 |
| Delicious (983 labels) | 7063 | 8015 | 7541 | 6911 | 3956 | 1611 |
| **Multitask regression** | | | | | | |
| RF1 (8 tasks) | 16102 | 17076 | 16815 | 17001 | 19999 | 19994 |
| SCM20D (16 tasks) | 19992 | 19991 | 19993 | 19992 | 19998 | 19998 |

## 2.6 Comparison with GBDT-MO

Here we provide details on comparison of SketchBoost and CatBoost with GBDT-MO and GBDT-MO (sparse) introduced in [Zhang and Jung, 2021]. The GBDT-MO implementation[4] is evaluated on CPU utilizing 8 threads per run (as it was done before). The experiment design and datasets are taken from original paper [Zhang and Jung, 2021]. For all the evaluated algorithms, we use hyperparameters provided in the original paper. For GBDT-MO, we use the best sparsity parameter $K$ which is also provided in the original paper. The only difference in our experiments is model training and evaluation which is done using 5-fold cross-validation (see Section 2.2) instead of using the test set for both early stopping and performance evaluation (as was done in the experiments for GBDT-MO[5]). We argue the latter method leads to the effect of quality overestimation. We also note that in the original paper results for GBDT-MO (sparse) are provided only for 4 datasets out of 6 datasets considered (and we also use only these 4 datasets). As it is done in the original paper, we use accuracy as the performance measure. The experimental results are given below.

Table 10: Comparison with GBDT-MO. Test scores (accuracy for classification and RMSE for regression) and their standard deviation for all sketching dimensions $k$.

| Algorithm | | Dataset | | | |
| --- | --- | --- | --- | --- | --- |
| | | MNIST (10 classes) | Caltech (101 classes) | NUS-WIDE (81 labels) | MNIST-REG (24 tasks) |
| CatBoost | | 0.9684±0.004 | 0.5049±0.0167 | 0.9893±0.0001 | 0.2708±0.0023 |
| GBDT-MO Full | | 0.976±0.004 | 0.4469±0.059 | 0.9891±0.0002 | 0.2723±0.0026 |
| GBDT-MO (sparse) | | 0.9758±0.0048 | 0.4796±0.0375 | 0.9892±0.0006 | 0.2736±0.0017 |
| SketchBoost Full | | 0.973±0.0028 | 0.5549±0.008 | 0.9893±0.0002 | 0.266±0.0019 |
| Random Sampling | $k=1$ | 0.973±0.0045 | 0.5704±0.0273 | 0.9892±0.0003 | 0.2671±0.0011 |
| | $k=2$ | 0.975±0.0034 | 0.5704±0.0174 | 0.9891±0.0003 | 0.2678±0.0015 |
| | $k=5$ | 0.9755±0.0042 | 0.5599±0.0146 | 0.9887±0.0002 | 0.2671±0.0012 |
| | $k=10$ | 0.9753±0.0007 | 0.5623±0.0165 | 0.989±0.0002 | 0.2661±0.0019 |
| | $k=20$ | – | 0.5691±0.0127 | 0.9889±0.0001 | 0.2665±0.0014 |
| Random Projection | $k=1$ | 0.9737±0.0023 | 0.5623±0.0159 | 0.9897±0.0003 | 0.2657±0.0018 |
| | $k=2$ | 0.9722±0.0037 | 0.5537±0.0064 | 0.9893±0.0004 | 0.2661±0.002 |
| | $k=5$ | 0.974±0.0032 | 0.5605±0.0137 | 0.9896±0.0003 | 0.2658±0.0013 |
| | $k=10$ | 0.9722±0.0045 | 0.5358±0.0157 | 0.9893±0.0004 | 0.2663±0.0007 |
| | $k=20$ | – | 0.5488±0.0332 | 0.9892±0.0004 | 0.2654±0.0012 |

Table 11: Comparison with GBDT-MO. Training time per fold in seconds for all sketching dimensions $k$.

| Algorithm | | Dataset | | | |
| --- | --- | --- | --- | --- | --- |
| | | MNIST (10 classes) | Caltech (101 classes) | NUS-WIDE (81 labels) | MNIST-REG (24 tasks) |
| CatBoost | | 156 | 136 | 13857 | 964 |
| GBDT-MO Full | | 362 | 776 | 2606 | 210 |
| GBDT-MO (sparse) | | 399 | 1312 | 3660 | 163 |
| SketchBoost Full | | 46 | 13 | 87 | 90 |
| Random Sampling | $k=1$ | 66 | 15 | 36 | 110 |
| | $k=2$ | 99 | 42 | 145 | 85 |
| | $k=5$ | 102 | 40 | 148 | 98 |
| | $k=10$ | 88 | 41 | 151 | 120 |
| | $k=20$ | – | 40 | 158 | 78 |
| Random Projection | $k=1$ | 45 | 16 | 72 | 38 |
| | $k=2$ | 70 | 13 | 71 | 38 |
| | $k=5$ | 66 | 15 | 73 | 51 |
| | $k=10$ | 70 | 14 | 49 | 44 |
| | $k=20$ | – | 14 | 48 | 45 |

---

[4] https://github.com/zzd1992/GBDTMO
[5] https://github.com/zzd1992/GBDTMO-EX

## 2.7 Experiment with synthetic dataset

The aim of this experiment is to illustrate the dependence of the time cost for training 100 trees on the number of outputs for popular GBDT frameworks on GPU. To do this, we train each framework twice on each task for 100 and 200 iterations and then calculate the difference in time. This allows us to estimate the time costs of 100 boosting iterations regardless the constant time costs such as features quantization and data transfer.

In more detail, since our goal is not to measure model quality and since we need alike datasets that vary only in the output dimension, we consider synthetic datasets generated (with the same feature parameters) by the algorithm proposed in [Guyon, 2003] and implemented in the scikit-learn library (v1.0.2)[6]. The dataset features are generated with 2000k rows and 100 features (10 features ar informative, 20 features are their linear combinations, and others are redundant). At each iteration, the number of classes is changed over the grid $\{5, 10, 25, 50, 100, 250, 500\}$. After the dataset is generated, we compute and report the time difference between 100 and 200 iterations for XGBoost, CatBoost, and SketchBoost with Random Projections (sketch dimension $k = 5$). The hardware used in this experiment is the same as described in Section 2.2. The main hyperparameters are chosen to be similar for all boosting frameworks. Namely, (1) trees are grown with depth-wise policy with maximal depth limited to 6, (2) row and column sampling is disabled, (3) learning rate is set to 0.01, and (4) L2 regularization term is set to 1, L1 regularization is disabled.