# OpenReview forum: "SketchBoost: Fast Gradient Boosted Decision Tree for Multioutput Problems"
_NeurIPS.cc/2022/Conference — NeurIPS 2022 Accept_

### Official Review · Reviewer_1K6v · 2022-07-08

**Rating:** 7
**Confidence:** 3
**Soundness:** 3 good
**Presentation:** 3 good
**Contribution:** 3 good

**Summary:**

The authors present three approaches to improve Gradient Boosted Decision Trees (GBDT) for multi output problems. The main goal is to speed up the training time of GBDT especially for those datasets with high number of classes. The authors' approaches focuses on a more efficient way to compute the scoring function, which maintains similar levels of performance (predictive performance).
The authors approaches are not a substitute for XGBoost, CatBoost or any SOTA GBDT algorithm, but rather can be integrated with any of these solutions.
Finally, the authors present extensive experiments to validate their solution, showing significant decrease in training time without compromising predictive performance.


**Questions:**

The questions are part of the previous section (Strengths and Weaknesses).
But to summarize them
- Table 1, error metrics reported of running everything on GPU?
- Why in Table 4 SketchBoost Full is taking less training time?
- Why GBDT-MO Sparse, Full and CatBoost (Table 4) are on CPU?

**Limitations:**

I don't have anything to add here.

**Strengths And Weaknesses:**

I believe this paper is well-written, tackles a relevant problem, considering how often these kind of algorithms are used in industrial settings, has a solid theoretical background, which is then evaluated with extensive experiments on different datasets.

In particular, I believe the strengths of this paper are:
- Problem that they tackle is relevant
- The solution is built to be used on top of existing SOTA algorithms
- SOTA algorithms are also compared to their solution, showing promising results
- Nice description of the complexity analysis between their solution and having the standard approach (Section 3.4).
- Nice comparison by running the baselines also on the GPU
- Good and promising results overall.
- Code available

A few things to improve that are worth mentioning:
- Conclusions section: I am missing a conclusions section where you summarize the goal of the paper and the findings. It reads strange and like there is something missing when the paper just ends in the last comparison of the algorithms.
- Table 1: The error metrics reported are after running all algorithms in the GPU right? (same as for Table 2, that its mentioned explicitly).
- Would be good to clarify why in Table 4 SketchBoost Full is taking less time to run than Random Sampling and Random Projection? I was expecting that the approaches presented by the authors would take less training time, as occurred during the comparison with CatBoost and XGBoost (Table 2). Is there a reason for this?
- For the second experiment, the comparison against GBDT-MO sparse and full are conducted on the CPU. Same as for CatBoost. Is the reason for this because the implementations that are available for such algorithms are only for the CPU? It would be nice to clarify. Same as why in that case CatBoost is run on the CPU (I guess to make it comparable to the other two).
- A figure similar to Figure 1 but with the authors proposed algorithms would be nice to include (I haven't checked if it's part of the supplementary material)

---

> ### Author Response · Authors · 2022-08-02
> **Response to Reviewer 1K6v**
>
> We thank Reviewer 1K6v for the comprehensive review and the vote to accept the paper. We are also grateful for summarizing the strengths of our work, in particular our analytical and empirical contributions, as well as noting the significance of our results. Below we give responses to the comments and questions raised in the review.
>
> **Conclusions section: I am missing a conclusions section where you summarize the goal of the paper and the findings.**
>
> Thank you for this comment! We apologize for not having the Conclusion section in the submission and agree that it is important to summarize the findings at the end of the paper. We will add this section in the camera-ready version if the paper is accepted (and we have an additional content page).
>
> **Table 1: The error metrics reported are after running all algorithms in the GPU right? (same as for Table 2, that is mentioned explicitly).**
>
> Thank you for this question. Table 1 and Table 2 refer to the same experiment. But since the processing units do not considerably affect test errors, we decided not to include this information in Table 1 for readability purposes. We are sorry for not making this clear. If you feel that this information is needed in Table 1, we will be happy to add it in the next revision.
>
> **Would be good to clarify why in Table 4 SketchBoost Full is taking less time to run than Random Sampling and Random Projection?**
>
> Thank you for this important comment! The reason for this is the following. If the dataset is small, then each boosting iteration requires little time. Therefore, when a sketching strategy is used, the speed up for each boosting iteration may be insignificant (especially because of ineffective utilization of GPU). At the same time, the number of iterations needed to convergence may be greater, which may result in an increase of the overall training time. Exactly this happened on some datasets from [Zhang and Jung, 2021]. We will clarify it in the next revision.
>
> **Why GBDT-MO Sparse, Full and CatBoost (Table 4) are on CPU?**
>
> Thank you for bringing this up! You are right, the reason for this is that the GBDT-MO implementation from Zhang and Jung [2021] is available only for the CPU. And to make its performance comparable to another algorithm, we decided to run CatBoost also on CPU. This information is now added; see page 9.
>
> **A figure similar to Figure 1 but with the authors proposed algorithms would be nice to include (I haven't checked if it's part of the supplementary material)**
>
> Thank you for the great suggestion! We did not have such a figure with SketchBoost. It is now given in Section 2.7 in the Supplementary Material (for your convenience, it is available also at https://sites.google.com/view/sketchboost/). We will include it in Conclusion if the paper is accepted and we have an additional content page.

---

> > ### Comment · Reviewer_1K6v · 2022-08-05
> > **Response to rebuttal**
> >
> > I would like to thank the authors for their response and for clarifying the concerns and questions that were posted.
> > Nothing else to add on my side.

---

> > > ### Author Response · Authors · 2022-08-08
> > > **Response to Reviewer 1K6v**
> > >
> > > Thank you for your feedback and suggestions for improving our work, especially the experiment section!

---

### Official Review · Reviewer_B4Dg · 2022-07-10

**Rating:** 7
**Confidence:** 4
**Soundness:** 3 good
**Presentation:** 3 good
**Contribution:** 3 good

**Summary:**

This paper aims to speed up the search process of the tree split in the training of Gradient Boosted Decision Tree (GBDT), especially for tasks with highly dimensional output. The authors propose three methods, Top-Outputs, Random Sampling and Random Projections,  to reduce the computational complexity of the split scoring function. Extensive experiments demonstrate the effectiveness of SketchBoost in terms of performance and efficiency.

**Questions:**

[Q1]: Random Projection appears to routinely outperform the other two methods; under what circumstances would you advise using the other two methods? It seems that the Random Projections is an advanced version of the other two proposed methods and the paper can be improved if the random matrix used in projection can be adaptive to the task datasets.

[Q2]: The sensitivity analysis of sketch dimension k indicates that reducing dimensions can result in distinct performance patterns. A interesting discovery is that Random Projections perform optimally when k = 1. When use SketchBoost for datasets with varying output dimensions, will there be a recommendation for selecting k?


**Strengths And Weaknesses:**

Pros:

[P1]: The paper is well written and organized in terms of clarity.

[P2]: Extensive tests are conducted to demonstrate the efficacy of the proposed strategy. Test results demonstrate that SketchBoost can obtain results equivalent to or even superior to existing SOTA approaches. SketchBoost is an order of magnitude faster on multidimensional datasets such as Dionis (355 classes) and Delicious (983 labels).

Cons:

[C1]: In order to deal with multi-dimensional tasks, some baseline could be considered. ( e.g LightGBM according to [1], a Deep learning-based solution[2][3])

[C2]: A related discussion connecting to deep learning method would be better.

[C3]: Some typo like line 110/257.


[1] Deep Neural Networks and Tabular Data: A Survey, https://arxiv.org/abs/2110.01889

[2]VIME: Extending the Success of Self- and Semi-supervised Learning to Tabular Domain https://proceedings.neurips.cc/paper/2020/file/7d97667a3e056acab9aaf653807b4a03-Paper.pdf

[3] TabNet, https://arxiv.org/abs/1908.07442

---

> ### Author Response · Authors · 2022-08-02
> **Response to Reviewer B4Dg**
>
> We thank Reviewer B4Dg for the thorough review and the vote to accept the paper. We find it encouraging that our empirical results sound convincing and are appreciated by the reviewer. Below we give answers to the comments and questions raised in the review.
>
> **Q1: Random Projection appears to routinely outperform the other two methods … the paper can be improved if the random matrix used in projection can be adaptive...**
>
> Great question! Certainly, Random Projection outperforms Top Outputs and Random Sampling on the vast majority of datasets. However, there are cases where Random Projection shows slightly worse performance than other approaches; see, e.g., the results for Delicious in Figure 1 in the Supplement. Therefore, if one has sufficient resources and model performance plays an important role, we would recommend testing all three methods. If the resources are limited, according to our numerical study, it is better to use Random Projection.
>
> Regarding the adaptive projection, we hope that we have understood this comment correctly and what is written below answers the question. We recommend a predefined value k=5 (please see our response to Reviewer psGW). One can choose k as a fraction of the output dimension, but our experiments show that in general there will not be much difference in the performance. If your question was about choosing k adaptively at each boosting iteration, then it is a challenging open problem and we do not have a good solution yet. We had an idea to choose the sketch dimension using the error bounds given in Section 1 in the Supplement. However, it is very time-consuming since, to estimate the error at each boosting iteration, one needs to compute singular values of the gradient matrix (which has a quadratic complexity in the output dimension).
>
> **Q2: The sensitivity analysis of sketch dimension k indicates that reducing dimensions can result in distinct performance patterns... will there be a recommendation for selecting k?**
>
> Thank you for this comment! Reducing the sketch size certainly can result in distinct performance patterns. Loosely speaking, our methods work similarly to regularization. Depending on the dataset, different values of the sketch size k may be optimal. For example, Figure 2 (in the main text) shows that k=1 is optimal for Random Projections on Dionis, but on SF-Crime or MoA, k=20 performs better. The positive side of our experiments is that our methods work well for a wide range of values of k, which means that one can take simply k=5. However, it is also possible to add k to hyperparameters that are tuned. In our view, k will not play a significant role here taking into account how many hyperparameters boosting frameworks have and that hyperparameter optimization is usually done using the random search or Bayesian optimization.
>
> **C1 and C2: ... some baseline could be considered (e.g. LightGBM according to [1], a Deep learning-based solution [2][3])... related discussion connecting to deep learning methods would be better.**
>
> Thank you for bringing this question up since it allows us to emphasize that any improvement in GBDT is of high practical interest. The recent surveys ([1,5]) that discuss what solution for tabular data is better, neural networks or GBDTs, conclude “that algorithms based on gradient-boosted tree ensembles still mostly outperform deep learning models on supervised learning tasks”. Hence GBDT is still one of the most powerful tools for solving problems with tabular data.
>
> We have not compared SketchBoost to Neural Networks because it is not common to use DL approaches as baselines in the GBDT literature and an exhaustive comparison between existing DL approaches and GBDTs deserves its own investigation and is worthy of future work. Nevertheless, we liked the idea of having a DL-based solution in the experiments, so we added TabNet [3] as a baseline. The results and experiment details are available in the Supplement, Section 2.6 (also at https://sites.google.com/view/sketchboost/ for your convenience). They confirm the conclusion made in the surveys — GBDTs outperform TabNet.
>
> Regarding LightGBM, the reason why we have not considered it as a baseline is that it uses the same multiouput strategy as XGBoost (one-vs-all), performs pretty similar to XGBoost (see, e.g., Table 5 in [1]), and is not so efficient on GPU as XGBoost (see, e.g., [4]).
>
> *References*
>
> [1] Deep Neural Networks and Tabular Data: A Survey.
> [2] VIME: Extending the Success of Self- and Semi-supervised Learning to Tabular Domain.
> [3] TabNet: Attentive Interpretable Tabular Learning.
> [4] Benchmarking and Optimization of Gradient Boosting Decision Tree Algorithms.
> [5] Are Neural Rankers still Outperformed by Gradient Boosted Decision Trees?
>
> —
>
> Thank you for your questions and important comments! We will add as much of our discussion as is possible in the camera-ready version if the paper is accepted (and we have an additional content page).

---

> > ### Comment · Reviewer_B4Dg · 2022-08-03
> > **Response to rebuttal**
> >
> > Thank you for your responses,
> >
> > The vast majority of my concerns have been addressed thanks to your detailed responses. Given the actual impact of the GBDT-based multioutput scenario, I would prefer my score for the paper to remain unchanged. Looking forward to SketchBoost's open source.
> >
> > I'm open to the opinions of other reviewers.

---

> > > ### Author Response · Authors · 2022-08-08
> > > **Response to Reviewer B4Dg**
> > >
> > > Thank you for your feedback and for the idea to compare SketchBoost with TabNet!

---

### Official Review · Reviewer_psGW · 2022-07-12

**Rating:** 5
**Confidence:** 3
**Soundness:** 3 good
**Presentation:** 4 excellent
**Contribution:** 3 good

**Summary:**

The submission presents three methods to speed up split finding in multi-output gradient boosted decision tree ensembles. The best-performing method applies Gaussian random projections to compress the gradient matrix before splits are found based on the compressed matrix. Empirical results on several multi-class, multi-label, and multi-target regression problems show that this yields much faster training times while maintaining a similar level of accuracy.


**Questions:**

N/A

**Limitations:**

There does not seem to be an explicit discussion of limitations.

**Strengths And Weaknesses:**

What is proposed makes sense, and the paper is well written. The empirical results are presented "for the best k", which determines the amount of compression, and this seems to imply parameter tuning on the test set, which would be problematic. However, it seems doubtful that the overall findings would be affected significantly by this.

What is of greater concern is that the proposed method should probably be compared to other, more generic approaches to multi-target prediction that are based on compressing the vector of targets, e.g.,

Tsoumakas, G., Spyromitros-Xioufis, E., Vrekou, A., & Vlahavas, I. (2014, September). Multi-target regression via random linear target combinations. In Joint european conference on machine learning and knowledge discovery in databases (pp. 225-240). Springer, Berlin, Heidelberg.

Hsu, Daniel J., Sham M. Kakade, John Langford, and Tong Zhang. "Multi-label prediction via compressed sensing." Advances in neural information processing systems 22 (2009).

Joly, A., Geurts, P., & Wehenkel, L. (2014, September). Random forests with random projections of the output space for high dimensional multi-label classification. In Joint European conference on machine learning and knowledge discovery in databases (pp. 607-622). Springer, Berlin, Heidelberg.

Kapoor, A., Viswanathan, R., & Jain, P. (2012). Multilabel classification using bayesian compressed sensing. Advances in neural information processing systems, 25.

Cissé, M., Artieres, T., & Gallinari, P. (2012, September). Learning compact class codes for fast inference in large multi class classification. In Joint European Conference on Machine Learning and Knowledge Discovery in Databases (pp. 506-520). Springer, Berlin, Heidelberg.

Wicker, J., Tyukin, A., & Kramer, S. (2016, April). A nonlinear label compression and transformation method for multi-label classification using autoencoders. In Pacific-Asia Conference on Knowledge Discovery and Data Mining (pp. 328-340). Springer, Cham.

The choice of x axis in Figure 1, particularly given the small size of the font used to label the axis, may mislead the reader into thinking that dependence on the number of targets is not linear. Also, Figure 1 illustrates performance for multi-class classification. It seems multi-label classification would be a more obvious example. Finally, it is necessary to (somewhere) present details of the synthetic data and the exact configuration of the learning algorithms.

Typos:

There is a broken reference on Line 110.

"on the most multiclass datasets"

"it achieve"

"it is order of magnitude faster"

---

> ### Author Response · Authors · 2022-08-02
> **Response to Reviewer psGW**
>
> We thank Reviewer psGW for the thorough review and for noting the soundness of our results and the good presentation of the paper. Below we give detailed responses to the main concerns raised in the review.
>
> **The empirical results are presented "for the best k" … and this seems to imply parameter tuning on the test set**
>
> Thank you for this comment. We regret that our presentation gives the impression that we propose to imply parameter tuning on the test set. Some of the empirical results are given “for the best k” only for the reason to summarize many experimental data into simple tables. Therefore, we would like to emphasize that (1) choosing k does not require iteration through a grid or using the test set, (2) the final performance of SketchBoost does not vary much in k; it can be seen, e.g, in Figure 2, (3) results for all values of k are given in the Supplement.
>
> Regarding the value of k, in practice we recommend using a predefined value k=5. This choice is based on our numerical study which shows that there is a wide range of values of k for which our methods work well. It is common in GBDT: modern toolkits have more than 100 hyperparameters, and most of them are not usually tuned (default values typically work well). Nevertheless, it is also possible to add k to parameters that are tuned (on the validation set, of course). In our view, an additional hyperparameter will not play a significant role here taking into account that hyperparameter optimization is usually done using the random search or Bayesian optimization.
>
> We will clarify this in the revision.
>
> **… the proposed method should probably be compared to other, more generic approaches**
>
> Thank you for bringing this topic up as it allows us to specify the place of our work in the literature. Thank you also for the suggested references, we will include them and the discussion below to the Related work section.
>
> Let us emphasize first that none of the approaches from the references combines the following four advantages of SketchBoost: (1) it is applicable to GBDT, (2) it speeds up the training process and does not drop down the quality, (3) it allows one to use any loss function, and (4) it does not rely on any specific data assumptions (e.g., sparsity or class hierarchy) or the problem structure (e.g., multi-label or multi-class).
>
> In more detail, existing approaches to multi-target problems usually fall into one of the following two categories: algorithm adaptation (AA) or problem transformation (PT). PT approaches reduce the number of targets using some compression techniques. Most of the mentioned papers ([2,4,5,6]) fall into this category and mainly differ in the choice of compression and decompression techniques. They pay a price in terms of prediction accuracy due to the loss of information during the compression phase. As a result, they do not consistently outperform the full baseline (e.g., see Section 2 in [7]). The PT methods are also not generic in the sense that they significantly rely on the problem structure or data assumptions. E.g., the method from [5] does not generalize directly to multi-label classification, the methods from [2,4] — to multi-task regression. The methods based on compressed sensing ([2,4]) can be applied to dense data, but this will affect the performance. Finally, PT methods usually decode the predictions directly to 0/1 but not to probabilities which are required sometimes in practice.
>
> Most PT approaches can be easily combined with SketchBoost. Namely, one can (1) make an encoding step beforehand to utilize sparsity or label similarity, (2) apply SketchBoost to a dense dataset, and (3) decode SketchBoost’s outputs to obtain the final predictions. Moreover, one can play with compression level in the encoding step and sketch dimension in SketchBoost to obtain the best possible performance.
>
> The only paper devoted to AA is [3]. The authors there use random projections to speed up random forests in the context of multi-label classification. The main difference from our work is that we consider GBDT and have no limitations on task type or loss function.
>
> Finally, the authors of [1] consider multi-target regression and construct new targets using random projections of existing ones. This idea is orthogonal to ours since it increases the number of targets (in order to improve performance of an algorithm).
>
> *References*
> [1] Multi-target regression via random linear target combinations.
> [2] Multi-label prediction via compressed sensing.
> [3] Random forests with random projections of the output space for high dimensional multi-label classification.
> [4] Multilabel classification using bayesian compressed sensing.
> [5] Learning compact class codes for fast inference in large multi class classification.
> [6] A nonlinear label compression and transformation method for multi-label classification using autoencoders.
> [7] FastXML: A fast, accurate and stable tree-classifier for extreme multi-label learning.

---

> > ### Author Response · Authors · 2022-08-08
> > **Response to Reviewer psGW**
> >
> > Dear Reviewer psGW,
> >
> > The reviewer-author discussion period will end on August 9. Have you had a chance to look at our answers yet? If you have any questions, we will be happy to answer them.
> >
> > We also would like to bring your attention to the fact that we made the font size in Figure 1 bigger as you suggested and added details on this experiment to the Supplementary Material; see Section 2.7. Also, the typos are now fixed, thank you for noticing them.
> >
> > Best regards,
> > The Authors

---

### Author Response · Authors · 2022-08-02
**Thank you very much for your reviews!**

We would like to thank the Reviewers for carefully reading our paper and writing thorough reviews. We are pleased that there is no doubting that Gradient Boosting is widely-used and any improvement of this algorithm is of high practical interest. We are also delighted that reviewers find our work theoretically sound and appreciate its significance.

We would like to take this opportunity to emphasize that none of the big GBDT implementations properly support multi-output problems, and we hope that our work can positively affect this situation.

Below we respond to each reviewer separately. Please let us know if you have additional questions or comments.

---

### Meta-Review · Area_Chair_CvFM · 2022-08-23

**Recommendation:** Accept
**Confidence:** Certain

**Metareview:**

After rebuttal, the reviewers unanimously agree that the submission should be accepted for publication at NeurIPS. Reviewers were excited about the achieved speed-up.

**Award:**

No

---

### Decision · Program_Chairs · 2022-09-14

Accept